EMBO
Molecular Medicine

# Dietary supplementation of arachidonic acid promotes humoral immunity

Shengyong Feng[1,2,12], Enhao Ma[1,12], Xiaona Na[3,4,12], Zongmei Wang[5,12], Wanbo Tai [2], Xinhui Bao[1], Mao Wang[1], Han Chang [6], Baolei Wu[3,4], Miaoxi Liu [7], Juzhen Li[1], Huicheng Shi[8], Celi Yang[3,4], Menglu Xi[3,4], Haibing Yang[3,4], Yuhan Li[1], Yibin Zhu[1,9], Penghua Wang [10], Ling Zhao [5 ✉], Ai Zhao [3,4 ✉] & Gong Cheng [1,2,9,11 ✉]

## Abstract

**Vaccination offers the most effective protection against contagious infectious diseases primarily by inducing humoral immunity. Vaccination efficacy is influenced by various factors. We report that dietary administration of arachidonic acid (ARA) significantly boosts rabies vaccine-induced production of neutralizing antibodies and protection against lethal rabies virus (RABV) infection in mice. In human volunteers, oral supplementation of ARA accelerates the expression of neutralizing antibodies to the levels sufficient for protection against RABV as early as one week after primary immunization. Mechanistically, ARA is enriched in lymph nodes and metabolized into immune modulators there. One of the ARA metabolites, prostaglandin $I_2$ ($PGI_2$), via the cyclic adenosine monophosphate (cAMP)-protein kinase A (PKA) axis, upregulates the expression of costimulatory molecule CD86, and activates activation-induced cytidine deaminase (AID) in B cells. These results suggest that ARA can be a potent dietary adjuvant to foster germinal center (GC) B cell response and humoral immunity.**

**Keywords** Germinal Center Response; Humoral Immunity; Arachidonic Acid
**Subject Categories** Evolution & Ecology; Immunology

## Introduction

Vaccines are the cornerstone of combating infectious diseases and act as the predominant means to defuse pandemic and epidemic risks, since they provide direct protection to immunized individuals by pre-training the adaptive immunity (Largeron et al, 2015; Stevens and Bryant, 2023). Most vaccines against infectious diseases confer protection mainly through the induction of neutralizing antibodies against invading pathogens(Burton, 2002; Gruell et al, 2022); The optimal vaccination strategy for worldwide prevention would be a single-dose administration that achieves a full seroconversion rate soon. However, due to the suboptimal response, the majority of vaccination schedules incorporate a minimum of two injections, typically spaced a few weeks apart (Barbier et al, 2022; Morefield et al, 2005). This long-time window of vulnerability is fine with routine immunizations, but it becomes problematic in an emergency such as the COVID-19 pandemic when rapid induction of protective immunity is essential (Lucas et al, 2021; Sabbe and Vandermeulen, 2016). In addition, high-dose vaccines are also adopted as a measure to enhance the immune response (Grohskopf et al, 2020), which may increase the risk of pain, myalgia, fever and other antigen-related, local and systemic side effects (Couch et al, 2007; DiazGranados et al, 2014; DiazGranados et al, 2016), and also heighten the cost of immunization due to vaccine antigens are expensive to manufacture (Tregoning et al, 2018). Nonetheless, some vaccines have still not achieved the desired effect (Ramanathan et al, 2018; Tregoning et al, 2018).

Vaccines induce the maturation of antigen-activated B cells and undergo several processes such as isotype switching and somatic hypermutation in the GCs, a transient lymphoid structure in which naive B cells are activated to diversify into mature plasma cells (PCs) or memory B cells (MBCs) (Vinuesa et al, 2016). This enables the production of neutralizing antibodies with high affinity (Lederer et al, 2022; Lederer et al, 2020; Sharma et al, 2024; Victora and Nussenzweig, 2022). Thus, there is an urgent need to develop a safer strategy for the prompt maturation of antigen-activated B cells in GCs, enabling rapid and robust generation of neutralizing antibodies for effectively countering invading pathogens.

[1]New Cornerstone Science Laboratory, Tsinghua University-Peking University Joint Center for Life Sciences, School of Basic Medical Sciences, Tsinghua University, 100084 Beijing, China. [2]Institute of Infectious Diseases, Shenzhen Bay Laboratory, 518000 Shenzhen, China. [3]Vanke School of Public Health, Tsinghua University, 100084 Beijing, China. [4]Institute for Healthy China, Tsinghua University, 100084 Beijing, China. [5]National Key Laboratory of Agricultural Microbiology, Huazhong Agricultural University, Hubei Hongshan Laboratory, Key Laboratory of Preventive Veterinary Medicine of Hubei Province, Huazhong Agricultural University, 430070 Wuhan, China. [6]College of Life Science and Technology, Beijing University of Chemical Technology, 100029 Beijing, China. [7]Department of Microbiology & Immunology, University of California, San Francisco, CA 94110, USA. [8]College of Food Science and Light Industry, Nanjing Tech University, 211816 Nanjing, China. [9]Institute of Pathogenic Organisms, Shenzhen Center for Disease Control and Prevention, 518055 Shenzhen, China. [10]Department of Immunology, School of Medicine, University of Connecticut Health Center, Farmington, CT 06030, USA. [11]Southwest United Graduate School, 650092 Kunming, China. [12]These authors contributed equally: Shengyong Feng, Enhao Ma, Xiaona Na, Zongmei Wang. ✉E-mail: lingzhao@mail.hzau.edu.cn; aizhao18@mail.tsinghua.edu.cn; gongcheng@mail.tsinghua.edu.cn

Polyunsaturated fatty acids (PUFAs) are distinguished by the presence of multiple double bonds in their carbon chain and can be broadly categorized into Omega-3 (n-3) and Omega-6 (n-6) fatty acids (Liput et al, 2021). PUFAs are of paramount importance in human nutrition as they cannot be internally synthesized, making their dietary intake crucial for maintaining optimal physiological functions (Heird and Lapillonne, 2005; Smit et al, 2004). Recently, there has been a growing interest in the roles of PUFAs as crucial modulators of the human immune system (Miles et al, 2021). Arachidonic acid (ARA) as an n-6 PUFA plays an important role in the proliferation of immune cells and the overall inflammatory responses, either directly or via its oxidized derivatives collectively known as eicosanoids (Yui et al, 2015). Thus, the immunomodulatory properties of PUFAs have driven investigations into their potential as immune therapeutics. Understanding the physiological functions and mechanisms of PUFAs in the immune system holds promise for developing dietary interventions and innovative therapeutic strategies to regulate immune responses. In this study, we demonstrate that supplementation of ARA, a food-based adjuvant, fosters the GC response in lymphoid tissues after vaccination, thereby accelerating and boosting the production of neutralizing antibodies in mice and humans.

# Results

## Supplementation of ARA enhances the production of antigen-specific antibodies in immunized mice

PUFAs play intricate functions in the regulation of immune responses. We therefore assessed the role of 6 PUFAs representing both n-3 and n-6 essential fatty acids for human health, including ARA, linoleic acid (LA), gamma-linolenic acid (GLA), EPA, DHA, and alpha-linolenic acid (ALA), in the generation of antigen-specific antibodies in immunized mice. To ensure the effective in vivo release of PUFAs and eliminate stress-induced physiological confounders (Almoshari, 2022; Patel et al, 2021), we exploited a micro-osmotic pump with a sustained-release capacity to assess the effect of these fatty acids in immunized animals. The BALB/c mice were subcutaneously implanted with a PUFA-releasing osmotic pump or with a pump releasing phosphate-buffered saline (PBS) as a negative control. Three days later, the animals were immunized with 20 μg of chicken ovalbumin (OVA). Induction of OVA-specific antibodies was then determined weekly beginning from the $2^{nd}$ week post-immunization (Fig. 1A). The subcutaneous supplementation of ARA significantly enhanced the titers of serum anti-OVA immunoglobulin G (IgG), compared to that of control and other PUFAs (Fig. 1B). Consistently, the proportion and the numbers of OVA$^+$ B220$^{low}$ CD138$^+$ PCs were largely increased in the lymph nodes and spleen of immunized mice with ARA-releasing osmotic pumps (Fig. 1C,D). Furthermore, the number of OVA-specific antibody-secreting cells (ASCs) was dramatically enhanced in the lymph nodes of immunized mice with ARA-sustaining supplementation, measured by an enzyme-linked immunospot (ELISpot) assay (Fig. 1E). Liquid chromatography-tandem mass spectrometry (LC-MS/MS) profiling of murine plasma across experimental groups demonstrated that supplementation with each targeted PUFA significantly elevated its corresponding plasma concentration (Fig. EV1A), confirming that this dosage effectively evaluates fatty acid functionality. Notably, the ARA-supplemented group exhibited reduced LA levels without significantly altering the levels of other fatty acids. Supplementation with other fatty acids primarily increased their concentrations without substantial cross-impact (Fig. EV1A).

PUFAs are essential fatty acids procured from dietary sources. Dietary surveys have revealed considerable variation in personal ARA intake across different regions and dietary habits, ranging from approximately 100 mg/day in developed countries to a mere 39 mg/day in the low-income countries (Forsyth et al, 2017; Kawashima, 2019). We therefore assessed whether dietary supplementation of ARA may enhance the production of antigen-specific antibodies after immunization. A serial amount of ARA (0 mg, 1.25 mg, 5 mg) was orally administered to the BALB/c mice every day. Other PUFAs, DHA and LA, were supplemented as unrelated controls to evaluate the effectiveness of ARA further. Three days later, the ARA-fed animals were immunized with OVA, and then the immunized mice were further administered daily with an equal amount of ARA, DHA, or LA for an additional 7 days. The serum antigen-specific antibodies were determined at 2 weeks, 3 weeks, and 4 weeks post-immunization. Consistently, the ARA intake through dietary supplementation largely fostered the production of anti-OVA antibody in a dose-dependent manner (Fig. EV1B). We next investigated the impact of the timing of ARA pre-administration on the immunization efficacy. Five mg of ARA was pre-administered daily to mice by oral gavage for various days (0, 3, 7) before immunization and continued for 7 days after immunization. The effect of either 3-day or 7-day ARA pretreatment on the OVA-antibody production (ARA-d −3/−7) was better than that without pretreatment (ARA-d 0). Nonetheless, the 3-day and 7-day ARA pretreatment regimens showed a similar effect (Fig. EV1C). In addition, the ARA pretreatment regimen was safe to animals, evidenced by changes in body weight, normal levels of alanine aminotransferase (ALT) and aspartate aminotransferase (AST), as well as blood routine indexes (Fig. EV1D,E; Appendix Table S1). We therefore applied the 3-day ARA pretreatment (5 mg/day) to the rest of the experiments. In this experimental setting, dietary administration of ARA significantly enhanced the titers of anti-OVA IgG from 2 weeks throughout 24 weeks after vaccination compared to the dietary administration of PBS (Fig. 1F), demonstrating an impressively long-lasting effect of ARA. Overall, these results indicate that ARA is a unique PUFA that can potently promote the proliferation of antigen-specific B cells following immunization and thus production of antigen-specific antibodies.

We next investigated the mechanism by which ARA promotes the generation of antigen-specific B cells and antibodies. The development of antigen-specific antibody responses requires the interaction between T cells and antigen-presenting cells such as dendritic cells (DCs). DCs orchestrate the stimulation of antigen-specific T-helper cells, thus leading to the clonal expansion of activated antigen-specific B cells in GCs (Stebegg et al, 2018). The induction of helper T cells is intricately regulated by factors such as the DC subset or innate cytokines (Persson et al, 2013; Schmitz et al, 2005; Stebegg et al, 2018). In order to determine the effect of ARA on various lymphocytes in naive mice, we supplemented mice with ARA for 10 days and collected lymph nodes and spleen on day 3 and day 10 for flow cytometry analysis, respectively (Fig. EV2A). Results showed that daily oral administration of ARA did not change the percentage and the numbers of various lymphocytes in

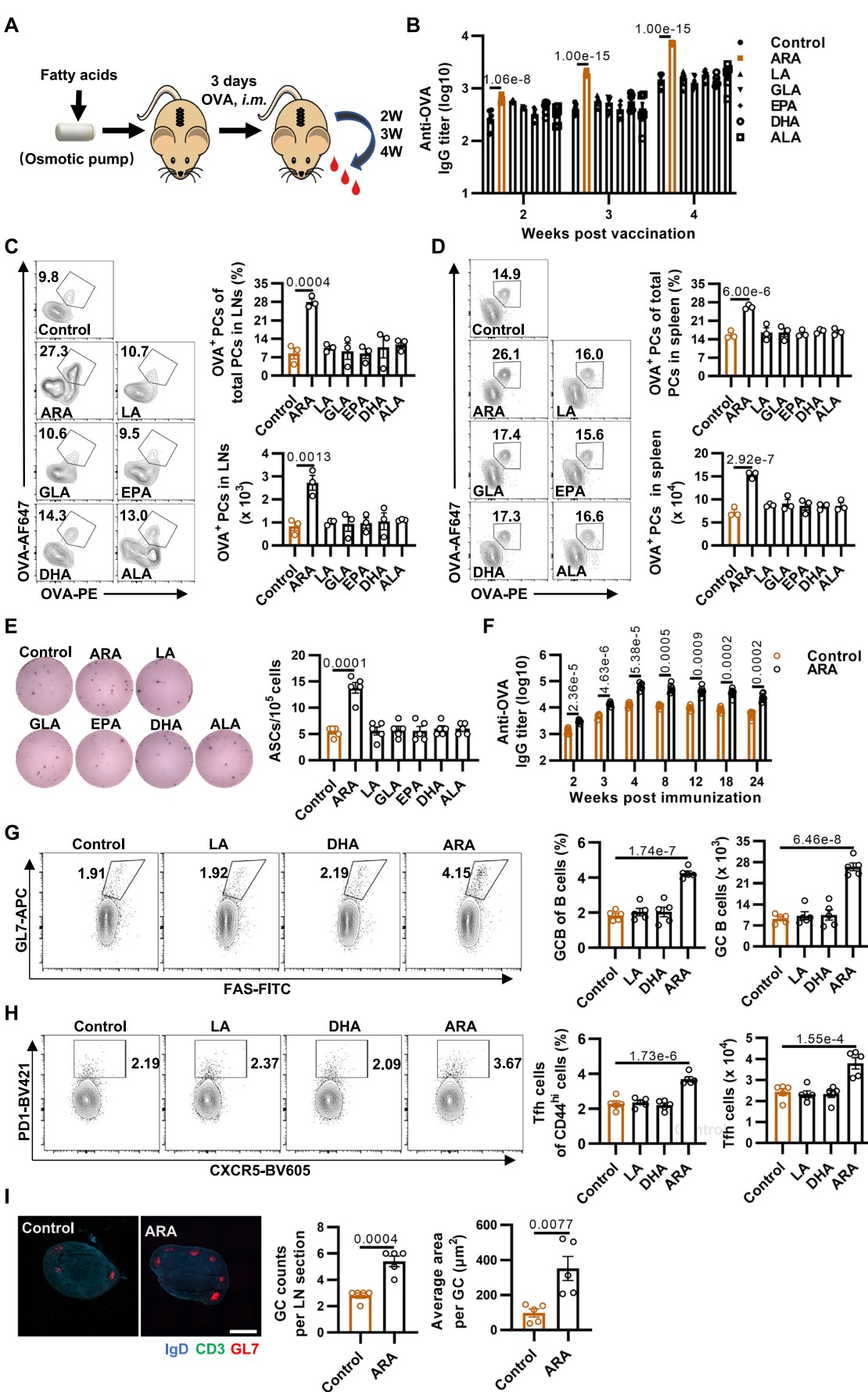

**Figure 1. Screening for long-chain polyunsaturated fatty acids (PUFAs) that promote antibody production.**

(A) Schematic diagram of the study design. An osmotic pump containing 10 mg PUFAs was implanted subcutaneously in mice (6–8 weeks BALB/c strain if not otherwise specified), and the mice were i.m. immunized with 20 µg OVA (with 500 µg alum, if not otherwise specified) 3 days later. Blood samples were collected on at 2 weeks, 3 weeks, and 4 weeks after i.m. immunization to measure the OVA-specific immunoglobulin (IgG) titers. (B) OVA-specific IgG titers of mice at 2 weeks, 3 weeks, and 4 weeks after i.m. immunization with OVA under subcutaneous pump supplementation of PUFAs ($n = 6$). (C, D) Flow cytometry analysis of PCs (OVA$^+$ B220$^{low}$ CD138$^+$) from draining lymph nodes (C) and spleen (D) at 3 weeks after i.m. immunization with OVA ($n = 3$). Left: Representative flow cytometry plots of OVA-specific PCs. Right: Statistical data of the percentages and cell numbers of OVA-specific PCs. (E) Representative photograph of ELISpot plate and quantification of OVA-specific IgG ASCs from draining lymph nodes at 3 weeks post-immunization ($n = 5$). (F) OVA-specific antibody titers from 2 weeks throughout 24 weeks after vaccination with OVA under an experimental setting that 3-day ARA pretreatment (5 mg/day) before immunization ($n = 6$). (G) Flow cytometry analysis of GC B cells (CD19$^+$ B220$^+$ GL7$^+$ Fas$^+$) on day 10 after immunization with OVA. Left: Representative flow cytometry plots of GC B cells. Right: Statistical data of the percentages and cell numbers of GC B cells ($n = 5$). (H) Flow cytometry analysis of Tfh cells (B220$^-$ CD3$^+$ CD8$^-$ CD4$^+$ CD44$^{hi}$ CXCR5$^+$ PD-1$^+$) in the draining lymph nodes on day 10 post-OVA immunization. Left: Representative flow cytometry plots of Tfh cells. Right: Statistical data of the percentages and cell numbers of Tfh cells ($n = 5$). (I) Representative images of whole lymph node sections. Lymph node sections were stained with monoclonal antibodies against IgD (blue), GL7 (red), CD3 (green). Scale bars were 500 µm. The left-hand plot shows GC numbers per lymph node, and the right-hand plot shows the average area of GC ($n = 5$). Data are representative of two or three independent experiments. All graphs represent mean ± SEM, and each data points represent individual mice or individual samples. Statistical significance was calculated by one-way ANOVA with Tukey's multiple comparisons test (B–E, G, H) and unpaired two-tailed $t$ test (F, I). Source data are available online for this figure.

the murine lymph nodes and spleen (Fig. EV2B–D), indicating that the daily ARA diet does not influence the immune homeostasis. In the OVA-immunized animals, the proportion and the numbers of classic/tissue-resident macrophages (Appendix Fig. S1A) and conventional/migratory DCs (Appendix Fig. S1B) were not regulated by the supplementation of ARA in the lymph nodes, suggesting that ARA is dispensable for antigen presentation. Meanwhile, splenocytes of mice immunized with OVA were subjected to restimulation with OVA peptides to detect the T cells secreting cytokines (Appendix Fig. S2A). We found that supplementation of ARA did not alter the proportion of T cells secreting cytokines (Appendix Fig. S2B). Luminex assay suggested that the concentrations of various cytokines in the serum of immunized mice remained unchanged (Appendix Fig. S2C), suggesting that ARA is not essential for cytokine production. We next assessed the role of ARA in the development of antigen-activated B cells in the lymph nodes of immunized mice. Dietary administration of ARA enhanced the frequency and the numbers of CD19$^+$B220$^+$GL7$^+$FAS$^+$ GC B cells (Fig. 1G) and B220$^-$ CD3$^+$CD8$^-$CD4$^+$CD44$^{hi}$CXCR5$^+$PD-1$^+$ follicular helper T (Tfh) cells (Fig. 1H) by flow cytometry. Bigger GCs were more frequently observed in the lymph nodes of mice that dietary administration of ARA measured by an immunofluorescence assay with GL7 staining (Fig. 1I). The results indicate that supplementation of ARA promotes the GC response.

## ARA-metabolized eicosanoid promotes the germinal center response via the cyclic adenosine monophosphate-protein kinase A axis

Lymph nodes are important peripheral lymphoid organs that produce antibodies. To determine whether ARA can be enriched in lymph nodes after supplementation, we performed LC-MS/MS analysis on the lipid extracts from the lymph nodes. The lymph nodes were collected from the mice 10 days after ARA supplementation (Fig. 2A). The tissues from the mice fed PBS served as negative controls. Data showed that the abundance of ARA was significantly enhanced in the lymph nodes of the mice with dietary administration of ARA (Fig. 2B).

Accumulating evidence indicates that ARA can be converted into eicosanoids, a family of bioactive lipid mediators encompassing prostaglandins, leukotrienes, and thromboxanes. These eicosanoids assume a central role in regulating inflammatory and immune

processes (Harizi et al, 2008; Turolo et al, 2021). By serving as a substrate for eicosanoid production, ARA actively contributes to the generation of signaling molecules that stimulate and regulate immune responses. This dual responsibility accentuates the intricate interplay between ARA and immune regulation (Tallima and El Ridi, 2018; Yui et al, 2015). We therefore determined the presence of the ARA-metabolized eicosanoids in the lymph nodes of animals with dietary administration of ARA by LC-MS/MS. The lymph nodes were collected from the mice at 7 days post-immunization, and the tissues from the mice fed PBS served as negative controls. A total of 25 ARA-derived eicosanoids were measured by targeted LC-MS/MS analysis, for which each eicosanoid was quantified by its commercially available standards (Appendix Table S2). The abundance of 6 eicosanoids, including 6-keto prostaglandin $F_{1\alpha}$ (6-keto-PGF$_{1\alpha}$, the stable derivative of PGI$_2$), prostaglandin $E_2$ (PGE$_2$), prostaglandin $F_{2\alpha}$ (PGF$_{2\alpha}$), prostaglandin $D_2$ (PGD$_2$), thromboxane $B_2$ (TxB$_2$, the stable metabolite of thromboxane $A_2$, TxA$_2$) and 11-hydroxy-5,8,12,14-eicosatetraenoic acid (11-HETE) were significantly enhanced in the lymph nodes of mice with dietary administration of ARA (Fig. 2C). However, the eicosanoid from sera remained unchanged (Appendix Table S3). To determine which eicosanoid promotes the humoral immunity, next, these upregulated eicosanoids were injected into the inguinal lymph nodes of animals as previously described (Andorko et al, 2014; Johansen and Kündig, 2014). Due to the unstable nature, the analogs of eicosanoids were used for animal administration (Gryglewski, 2008; Nishio et al, 1997; Sirén et al, 1985). Mice supplemented with ARA served as positive controls. The treated mice were then immunized with OVA. The antigen-specific antibody titers were determined at day 28 post-immunization. Based on an enzyme-linked immunosorbent assay (ELISA), we found that Beraprost, the PGI$_2$ analog, significantly promoted the production of OVA-specific antibodies at 4 weeks post-immunization, which was consistent with the effect of ARA (Fig. 2D).

To demonstrate that PGI$_2$ is mainly produced by ARA, we conducted LC-MS/MS analysis on lymph nodes from mice with dietary administration of various PUFAs, we found that supplementing ARA instead of other PUFAs significantly promoted the production of PGI$_2$ (Fig. EV3A). PGI$_2$ exerts its action through the PGI$_2$ receptor (PTGIR), which was upregulated in lymph nodes of mice supplemented with ARA (Fig. EV3B). When activated by PGI$_2$, PTGIR induces adenyl cyclase (AC), leading to increased intracellular cyclic adenosine monophosphate (cAMP) (Vane and

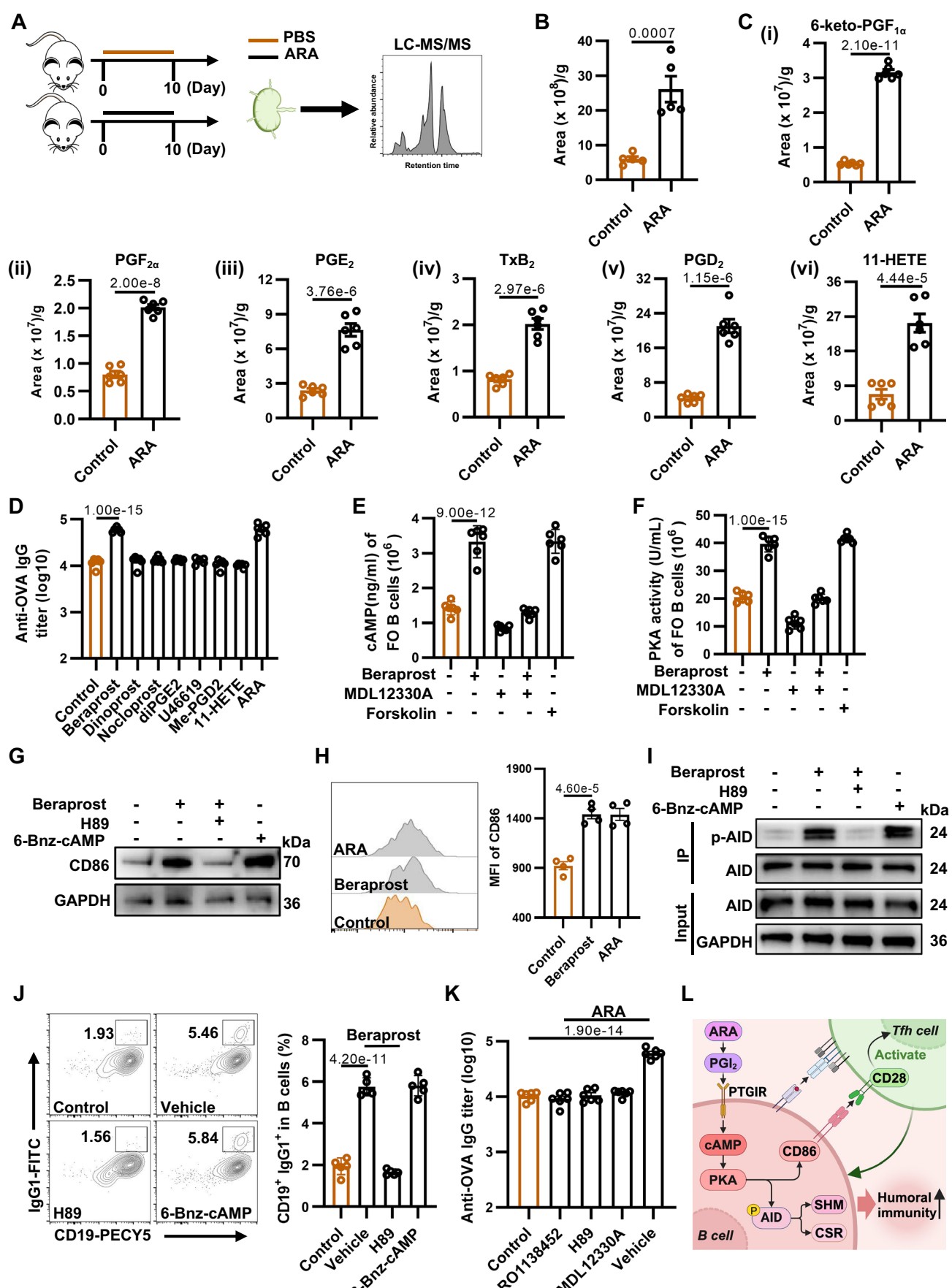

◄ **Figure 2.  ARA-metabolized eicosanoid promotes the germinal center response via the cAMP–PKA axis.**

(A) Schematic diagram of the study design. After supplementing ARA by gavage to mice for 10 days, lipids were extracted from the inguinal lymph nodes for liquid chromatography-tandem mass spectrometry (LC-MS/MS) analysis. (B) Quantification of ARA in the inguinal lymph nodes of the mice dietary administration of ARA by LC-MS/MS ($n = 5$). (C) Quantification of the significantly enhanced ARA-metabolized eicosanoids in mice's lymph nodes with dietary ARA administration by LC-MS/MS ($n = 6$). (D) OVA-specific IgG titers of mice supplemented with the upregulated analogs of the eicosanoids on day 28 after i.m. immunization with OVA ($n = 6$). Beraprost is the analog of $PGI_2$; Dinoprost is the analog of $PGF_{2\alpha}$; Nocloprost, a $PGE_2$ analog targeting EP1 and EP3; Di-$PGE_2$ (16,16-Dimethyl prostaglandin $E_2$) is a $PGE_2$ analog targeting EP2/EP4. U-46619 (9,11-Methanoepoxy $PGH_2$) is an analog of $TXA_2$; Me-$PGD_2$(15(R)-15-methyl $PGD_2$) is a metabolically stable analog of $PGD_2$. (E, F) The intracellular cAMP levels (E) and PKA activity (F) of FO B cells were stimulated with indicated compounds: Beraprost (500 nM); MDL12330A (20 µM), the adenylyl cyclase (AC) inhibitor; Forskolin (10 µM), the AC activator, was used as the positive control ($n = 6$). (G) The expression level of CD86 in FO B cells stimulated with indicated compounds by immunoblotting analysis. Beraprost (500 nM); H89 (10 µM), the PKA inhibitor; 6-Bnz-cAMP (200 µM), the PKA activator, was the positive control. (H) Mean fluorescence intensity (MFI) of CD86 on FO B cells from lymph nodes of mice 10 days post-immunization with OVA ($n = 4$). Left: Representative flow cytometry plots. Right: Statistical data of MFI of CD86. (I) The phosphorylation level of AID in FO B cells stimulated with indicated compounds by IP assay. Beraprost (500 nM); H89 (10 µM); 6-Bnz-cAMP (200 µM). (J) Flow cytometry analysis of IgG1 expression in activated murine B cells treated with the Beraprost or Beraprost and H89, under stimulation with LPS plus IL-4. 6-Bnz-cAMP (200 µM) was the positive control ($n = 5$). Left: Representative flow cytometry plots of $CD19^+$ $IgG1^+$ B cells. Right: Statistical data of the percentages of $CD19^+$ $IgG1^+$ B cells. (K) OVA-specific IgG titers of mice supplemented with ARA under various inhibitors treatment on day 28 after immunization with OVA. RO1138452, inhibitor of PTGIR. MDL12330A, inhibitor of adenyl cyclase. H89, inhibitor of protein kinase A ($n = 6$). (L) A mechanical scheme of supplementing ARA to promote humoral immunity. Data are representative of two or three independent experiments. All graphs represent mean ± SEM, and all data points represent individual mice or individual samples. Statistical significance was calculated by unpaired two-tailed *t* test (B) and one-way ANOVA with Tukey's multiple comparisons test (D–F, H, J, K). Source data are available online for this figure.

Corin, 2003). And the main intracellular target of cAMP in mammalian cells is cAMP-dependent protein kinase A (PKA) (Walsh et al, 1968). Furthermore, the primary function of follicular (FO) B cells is to facilitate the production of adaptive antibodies (Wang et al, 2020). Hence, we sorted FO B cells from the spleens of unprimed mice to investigate whether $PGI_2$ could trigger cAMP–PKA signaling pathways. Data showed that Beraprost significantly increased intracellular cAMP levels and PKA activity in FO B cells, which could be markedly abrogated by cis-N-(2-phenylcyclopentyl)-azacyclotridec-1-en-2-amine hydrochloride (MDL12330A), the AC inhibitor. As anticipated, forskolin, the AC activator, induced the activation of the cAMP–PKA signaling pathway (Fig. 2E,F).

CD86 (also known as B7-2), a key costimulatory molecule expressed on antigen-presenting cells, including B cells, is critical for GC formation and the Tfh phenotype maintenance (Borriello et al, 1997; Salek-Ardakani et al, 2011). A recent study confirms that CD86 can be induced through the cAMP–PKA axis in B cells (Wolf et al, 2022). To assess whether $PGI_2$ can upregulate CD86 expression via the cAMP–PKA axis, we treated FO B cells with Beraprost, anti-CD40, Interleukin-4 (IL-4) and anti-IgM. Both Western Blotting and flow cytometric analysis showed that CD86 was significantly upregulated, yet this upregulation was counteracted by treating with a PKA inhibitor, H89 (Figs. 2G and EV3C). We also found that Beraprost induced CD86 expression in a dose-dependent manner (Fig. EV3D). The results indicate that the expression of CD86 can be upregulated by $PGI_2$ through the cAMP–PKA axis. Next, we explored whether CD86 was upregulated in vivo. We supplemented the mice with either ARA or Beraprost, and the draining lymph node was monitored on 10 days post-immunization by flow cytometry analysis. Data showed that both ARA and Beraprost treatment led to the upregulation of CD86 in draining lymph node B cells (Fig. 2H).

Activation-induced cytidine deaminase (AID, which is encoded by *Aicda*), GC B cells- and activated B cells-specific deaminase, is essential for the humoral immune response since they orchestrate class switch recombination and somatic hypermutation (Chandra et al, 2015; Muramatsu et al, 2000). Notably, PKA-mediated phosphorylation is required for AID activity (Barreto et al, 2003; Pasqualucci et al, 2006). To detect whether $PGI_2$ increases the phosphorylation of AID, purified FO B cells from naive BALB/c

mice were cultured in the presence of lipopolysaccharides (LPS) with or without Beraprost. Data from the immunoblotting experiment showed that Beraprost obviously increased the phosphorylation of AID, whereas H89, a PKA inhibitor, reversed the effect (Fig. 2I). As expected, 6-Bnz-cAMP, a PKA activator, enhanced the phosphorylation of AID (Fig. 2I). The enhancement of AID activity by $PGI_2$ was further demonstrated in the class switch recombination induction system with LPS plus IL-4: Purified spleen B cells from naive mice were stimulated in the system for 4 days with or without Beraprost, and the percentage of $CD19^+$ $IgG1^+$ double-positive B cells were detected at the endpoint of the experiment by flow cytometry. Data showed that the double-positive B cells were significantly increased in the presence of Beraprost (Fig. 2J), and the effect was in a dose-dependent manner (Fig. EV3E). Collectively, these observations demonstrate that AID activity is enhanced by $PGI_2$ through the cAMP–PKA axis in B cells.

To further confirm the efficacy of $PGI_2$ in vivo, we supplemented the immunized mice with ARA while treating them with RO1138452, MDL12330A and H89, which are inhibitors targeting PTGIR, adenyl cyclase and PKA, respectively. Both GC B cells and OVA-specific antibodies were measured 10 days and 28 days after immunization, respectively. Flow cytometric analysis showed that the inhibitors counteracted the percentages and the absolute cell numbers of $GL7^+FAS^+$ GC B cells (Fig. EV3F). And the enhanced OVA-specific IgG titers induced by ARA were also counteracted by these inhibitors (Fig. 2K). To determine whether $PGI_2$ or ARA has an effect on the proliferation of B cells or T cells, we added Beraprost or ARA to sorted B cells or T cells in vitro. The results showed that the proliferation of B cells or T cells was not influenced in the presence of ARA or Beraprost (Fig. EV3G,H). Collectively, $PGI_2$, derived from ARA, promotes the humoral immune response by boosting the expression of CD86 and increasing the activity of AID via the cAMP–PKA axis in B cells (Fig. 2L).

## Dietary ARA enhances rabies vaccine-elicited humoral immunity in mice and humans

Vaccination efficacy is influenced by many factors, including but not limited to vaccine factors (such as immunogenicity, adjuvant or

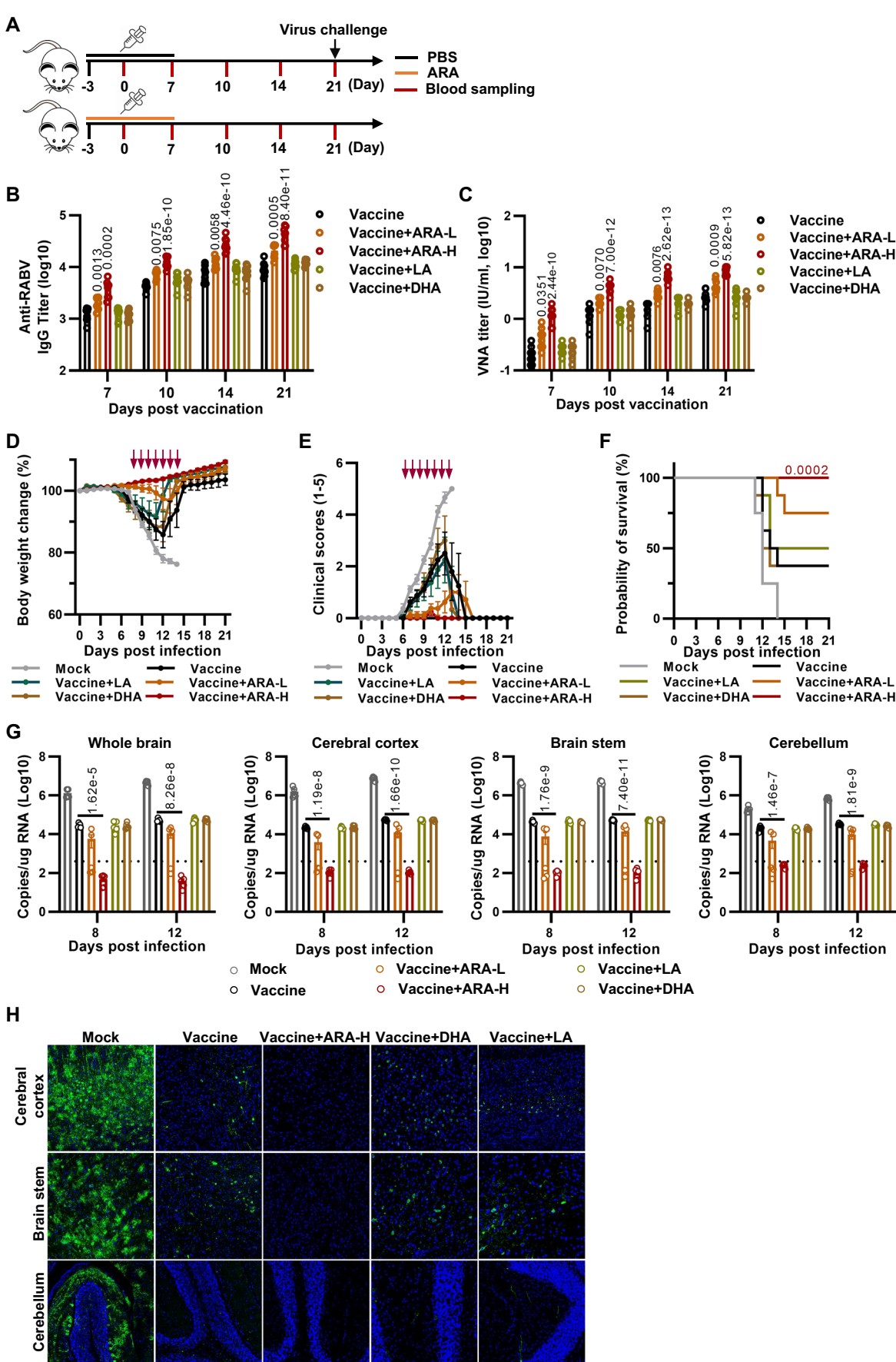

**Figure 3. ARA supplementation protects mice against virulent RABV challenge by enhancing humoral immunity.**

(A) Schematic diagram of the study design. BALB/c mice were orally administered 1.25 mg or 5 mg of ARA, 5 mg of DHA, 5 mg of LA, or PBS daily for 3 days in advance. Mice were immunized with 100 μL of the solution containing $10^7$ FFU of inactivated rabies vaccine on day 0 and administered different PUFAs or PBS orally daily for an additional 7 days. The serum was collected on days 7, 10, 14, and 21 after immunization. (B, C) The anti-RABV IgG titer (B) and RABV-specific VNA titer (C) in serum were measured by ELISA and fluorescent antibody virus neutralization (FAVN) assay, respectively ($n = 8$). Vaccine+ARA-L, supplementing mice with 1.25 mg ARA daily; Vaccine+ARA-H, supplementing mice with 5 mg ARA daily. (D–F) On day 21 post-vaccination, the mice were challenged by $100LD_{50}$ of RABV, and body weight changes (D), clinical scores (E), and survival ratios (F) were monitored daily for 21 days. Unvaccinated mice as mock group ($n = 8$). (G) In a parallel group of mice, the brains were collected on day 8 and 12 post-infection. The mRNA levels of RABV-N in the whole brain, cerebral cortex, brain stem, and cerebellum were analyzed by qPCR on days 8 and 12 post-infection ($n = 5$). (H) Immunofluorescence stain of cerebral cortex, brain stem and cerebellum. The brain on day 12 post-infection was stained with polyclonal antibody against RABV-P and AF488-conjugated goat anti-rabbit IgG (green), and the nucleus was stained with DAPI (blue) ($n = 3$): scale bars were 50 μm. Data are representative of two independent experiments. All graphs represent mean ± SEM, and all data points represent individual mice or individual samples. Significance was calculated by one-way ANOVA with Tukey's multiple comparisons test (B–E, G) or log rank (Mantel–Cox) test (F). The arrows (D, E) indicate a significant difference between the group of vaccine+ARA-H and vaccine. Source data are available online for this figure.

delivery method) or intrinsic host factors (such as age, genetic background, metabolic status or health condition). Traditional efforts to improve vaccination efficacy have been largely focused on vaccine factors. In this research, we asked whether dietary supplementation of ARA could improve vaccine effectiveness against infectious diseases. As proof-of-principle, we selected rabies because of its high lethality and lack of herd immunity. The hypothesis was first tested our hypothesis in mice. We treated mice daily with individual PUFAs or PBS control for 3 days; then, the administered animals were inoculated with a single-dose inactivated rabies vaccine and continued the PUFA/PBS treatment for 7 days (Fig. 3A). The mice treated with both vaccine and PBS are the PBS group and the mice treated with only PBS without vaccine are named as the mock group. The virus challenge experiment will be conducted 3 weeks after immunization. Prior to challenge, plasma concentrations of ARA, LA, and DHA reverted to baseline levels, and no significant differences in ARA, LA, or DHA levels were observed between groups by LC-MS/MS analysis (Fig. EV4A).

Consistent with the results from OVA immunization, dietary administration of ARA significantly enhanced the titers of rabies virus (RABV)-neutralizing antibodies (VNA) in a dose-dependent manner (Fig. 3B,C). However, neither DHA nor LA improved the VNA production (Fig. 3C). Subsequently, these vaccinated mice were challenged with 100 $LD_{50}$ RABV at 3 weeks post-vaccination. Both animal body weight and clinical scores were recorded daily after the infection. From day 5 post-infection, the unvaccinated animals (mock group) started to lose body weight (Fig. 3D) and developed neurological complications such as staggering and paralysis (Fig. 3E), and all succumbed to lethal RABV infection by day 14 (Fig. 3F). The vaccinated mice, which received DHA, LA, or PBS (control) presented similar clinical symptoms with ~50% mortality (Fig. 3D–F). Dietary supplementation of ARA rendered mice resistant to rabies in a dose-dependent manner. Five mg of ARA per day provided complete protection to the animals against lethal RABV infection (Fig. 3D–F). Consistently, there was no detectable viral RNA in the neural tissues, including the whole brain, cerebellum, cortex, and brain stem of ARA-treated- and -vaccinated animals (5 mg/day), compared to the higher viral RNA loads in the other non-ARA groups (Fig. 3G). Furthermore, focus-forming unit (FFU) assays was conducted to detect the virus titer in the mouse brain, and the results were consistent with the virus RNA copy numbers (Fig. EV4B). The results of immunofluorescence microscopy further confirmed that ARA supplementation significantly reduced the viral antigen load in different parts of the

brain (Fig. 3H). To further demonstrate that ARA supplementation protects mice against virulent RABV challenge by enhancing humoral immunity, we transferred the sera of immunized mice supplemented with ARA or PBS to naive mice one day before challenge with 100 $LD_{50}$ of RABV. The results indicated that the mice received sera from the mice treated with ARA were completely protected, whereas the mice received sera from the mice treated with PBS showed approximately 50% protection, which was comparable to the efficacy of vaccination alone (Fig. EV4C,D). Nevertheless, under the same supplementary conditions mentioned above, ARA did not recover the clinical symptoms and survival of unvaccinated mice at all without vaccination (Fig. EV4E,F). Altogether, these results demonstrate that dietary administration of ARA significantly enhances the humoral immune response of a single-dose rabies vaccine in mice.

Rabies vaccination is recommended for people who are at high risk, such as veterinarians and animal researchers, or bites by rabid dogs. Most people are rabies vaccine-naive and free of pre-existing rabies immunity. In addition, ARA is categorized as a food material and thus can be directly applied in human diets. We therefore assessed whether dietary administration of ARA may boost the rabies vaccine-elicited anti-RABV humoral immune response in humans. A randomized, triple-blinded, placebo-controlled trial was conducted in rabies vaccine-naive people (Fig. 4A). Finally, 14, 15, and 15 participants in each group completed the trial, respectively (flow chart is shown in Fig. EV5A). The mean (standard deviation [SD]) age of the 44 participants was 24.40 (2.59) years, and 17 (38.6%) of them were male. The demographic characteristics and lifestyles at baseline are shown in Appendix Table S4, and no significant differences were found across the three groups. Energy and macronutrient intakes in the period of washout (days −6 to −4), the middle of intervention (days 5–7), and the end of intervention (days 11–13) are shown in Appendix Table S5, and no significant differences were found. LC-MS/MS profiling of fatty acid composition in the serum of volunteers showed that plasma ARA level was significantly increased in the ARA supplementation group, other fatty acids remained unchanged after ARA supplementation, except for LA, which was decreased after ARA supplementation (Fig. EV5B).

Results show that the anti-RABV-specific IgG titers were significantly higher in both ARA groups than in the placebo group, regardless of whether ARA was given prior to vaccination or not (Fig. 4B,C). Moreover, both anti-RABV IgG and VNA titers were much higher in the Pre-ARA group than in the ARA group

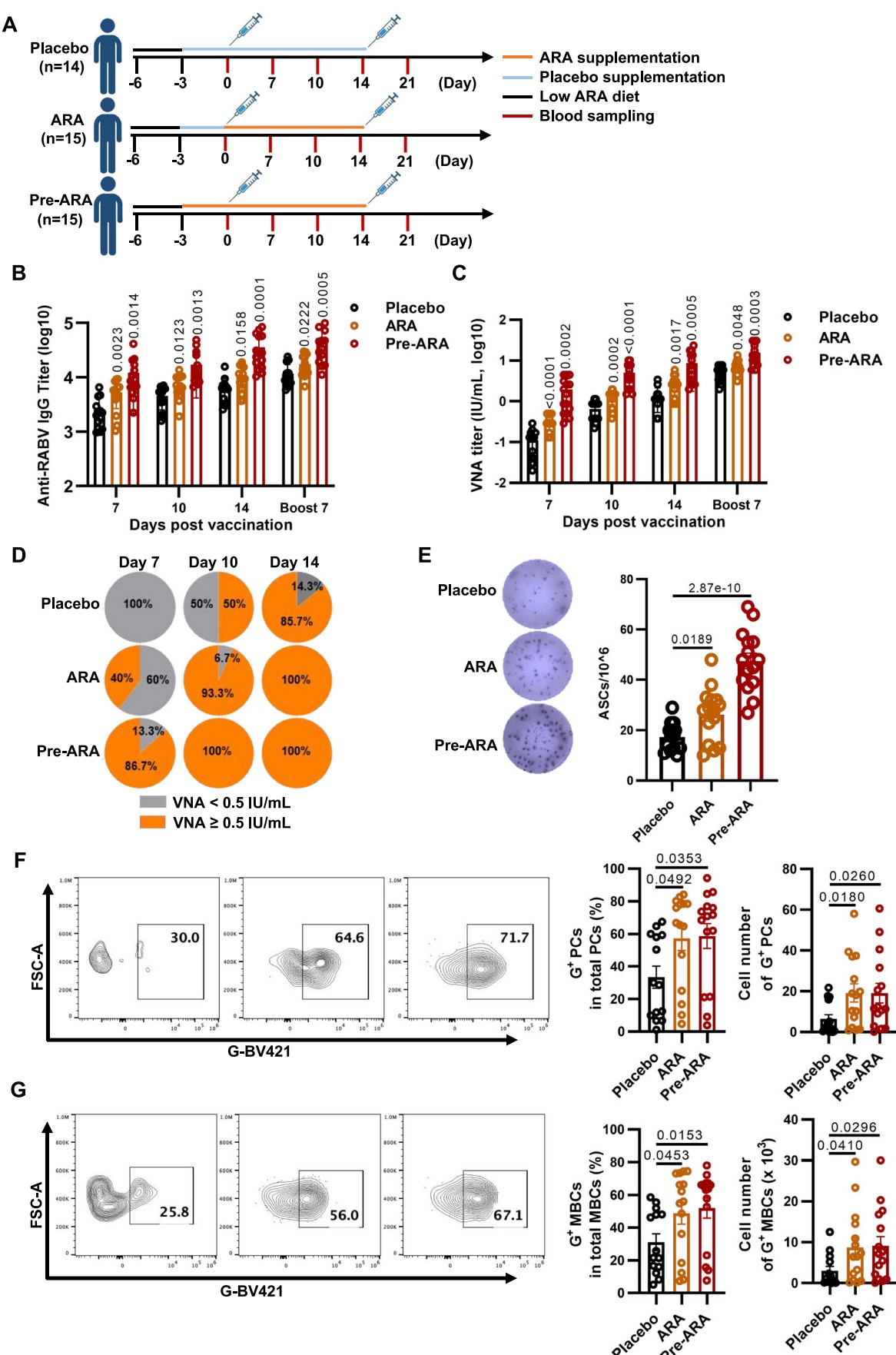

◄ **Figure 4.  Dietary ARA administration enhanced the anti-RABV humoral immune response in vaccinated humans.**

(**A**) Schematic diagram of the study design. This study included three periods: washout (days −6 to −4), supplementation (days −3/0–13), and post-supplementation (days 14–21). In the washout period, participants in all three groups were required to follow an ARA-restricted to minimize the individual variation of ARA intake at a baseline. The supplementation period had no further dietary restrictions. The placebo group received capsules containing sunflower seed oil daily (*n* = 14); the Pre-ARA group received capsules containing 512.4 mg of ARA daily from days −3 to 13 (*n* = 15) and the ARA group received the same supplementation as from day 0 to13 (*n* = 15). However, they received a placebo in the first 3 days of supplementation (days −3 to −1) to prevent unblinding. The RABV vaccine was injected on days 0 and 14. The blood samples were collected on days 0, 7, 10, 14, and 21. The serum was separated for the detection of rabies virus-specific antibodies and neutralizing antibodies. In addition, peripheral blood mononuclear cells were isolated for subsequent flow cytometry analysis. (**B, C**) The anti-RABV IgG titer (**B**) and RABV-specific VNA titer (**C**) in serum were measured by ELISA and FAVN assay (*n* = 14 in the Placebo group; *n* = 15 in ARA and Pre-ARA group, respectively). (**D**) The proportion of volunteers with seroconversion induced by ARA. VNA values greater than 0.5 IU/ml were considered positive. (**E**) Representative images of ELISpot assays. On the 13 days after the first shot vaccination, PBMCs were prepared and seeded, and then the RABV-specific ASCs were counted by an ELISpot assay (*n* = 14 in the Placebo group; *n* = 15 in ARA and Pre-ARA group, respectively). (**F, G**) Flow cytometry analysis of RABV-specific PCs (**F**) and RABV-specific MBCs (**G**) in PBMCs 13 days after the first shot vaccination. Left: Representative flow cytometry plots of RABV-specific PCs and RABV-specific MBCs in PBMCs. Right: Statistic data of the percentages and cell numbers of RABV-specific PCs and RABV-specific MBCs (*n* = 14 in the Placebo group; *n* = 15 in ARA and Pre-ARA group, respectively). All graphs represent mean ± SEM, and all data points represent individual volunteers. Significance was calculated by one-way ANOVA with Tukey's multiple comparisons test. Source data are available online for this figure.

(Fig. 4B,C), suggesting that administering ARA before vaccination is most effective. Next, we estimated whether these vaccination regimens provide full protection against rabies. According to recent studies, a VNA titer ≥0.5 IU/mL provides full protection (Zhang et al, 2023; Zhao et al, 2019). At 7 days after the first-shot vaccination, none of the vaccinees in the placebo group, yet 40% of vaccinees in the ARA group and 86.7% of vaccinees in the Pre-ARA group, gained full protection (Fig. 4D). On day 14 after the first-shot vaccination, all vaccinees from both ARA groups showed a level of anti-RABV VNA higher than 0.5 IU/mL, while 14.3% of vaccinees in the placebo still did not acquire the vaccine-mediated protection (Fig. 4D). To validate this finding further, we isolated the peripheral blood monocytes (PBMCs) from the blood of vaccinees. The vaccinees with ARA supplementation presented more ASCs (Fig. 4E), RABV-G-positive PCs and MBCs than the vaccinees with placebo (Fig. 4F,G). Other leukocytes from PBMCs, such as granulocytes, monocytes, CD4 T cells, and CD8 T cells, did not show significant changes (Fig. EV5C). These results exemplify the effectiveness of dietary ARA in boosting and accelerating humoral immune responses in humans.

Concentrations of ARA eicosanoids in plasma were further determined, and no significant difference was observed between the placebo group and the ARA supplementation groups (Appendix Table S6). No adverse effects related to the supplementation or vaccination were observed in any of the three groups (Appendix Table S7). It was found that the ARA supplementation did not change lipid, inflammation, or blood coagulation significantly, except that thromboplastin time (TT) changed differently across the three groups. However, after the intervention, the TT of all participants fell within the normal range.

## Discussion

Vaccines offer protection to individuals across diverse age groups, thereby underscoring their pivotal role in maintaining public health and global health security. The vaccine-mediated generation of protective antibodies is the principal mechanism of defense against various pathogens (Casadevall, 2018; Earle et al, 2021; Plotkin, 2010). The biological mechanisms underpinning the activation of antibodies and the associated GC responses are pivotal in the establishment of immunity via immunization. In this

study, we identified that supplementation of ARA, rather than other PUFAs, promoted the GC responses in lymphoid tissues of immunized hosts, thereby enabling maturation of the antibody-secreting PCs in the immediate early stage of immunization. Thus, the humoral immune responses can be rapidly and strongly boosted in the immunized animals with ARA supplementation. Mechanistic studies showed that the ARA can be metabolized into multiple eicosanoids in the lymphoid tissues. One metabolite, $PGI_2$, on the one hand, promotes the expression of CD86 through the cAMP–PKA axis and enhances the interactions between B cells and Tfh cells. On the other hand, it induces the activation of AID, and ultimately promotes the humoral immune response. Though LA can be converted to GLA and to ARA by stepwise desaturation and chain elongation, the conversion to ARA is very low (Emken et al, 1994; McCloy et al, 2004). Linoleic acid is readily oxidized by delta 6-desaturase to GLA. GLA elongation step to dihomo-c-linolenic acid (20:3-n6) is rapid; however, it is oxidized by delta-5 desaturase to yield ARA at a small percentage because delta-5 desaturase prefers the n-3 to n-6 fatty acids (Sprecher, 2002; Tallima and El Ridi, 2018; Wiktorowska-Owczarek et al, 2015), which may result in the antibody production not promoted by supplementation of LA and GLA. In addition, the distribution of delta-5 desaturase varies with tissues (Cho et al, 1999), which may also limit the conversion of LA and GLA to ARA.

ARA is converted to $PGH_2$ by the cyclooxygenase (COX) enzymes, and PGI synthase (PGIS) is the terminal enzyme that converts $PGH_2$ into $PGI_2$ (Lee et al, 2005). Previous studies have reported that PGIS is highly expressed by follicular dendritic cells (FDCs) in germinal centers, indicating that $PGI_2$ in lymph nodes is mainly produced by FDCs (Lee et al, 2005). Therefore, FDC can not only capture and retain antigens and present antigens to B cells (Suzuki et al, 2009; Wang et al, 2011) and support the survival and differentiation of B cells by providing cytokines (such as BAFF) and intercellular signals (such as Notch and Wnt signaling pathways) (Kim et al, 2012; Suzuki et al, 2010; Yoon et al, 2009), but also enhance the expression of CD86 and the activity of AID in B cells by producing $PGI_2$. Thus, FDCs plays an important role in the maturation and maintenance of humoral immune responses.

ARA is released from the membrane into the cell cytosol by Phospholipase $A_2$ ($PLA_2$) (Djuricic and Calder, 2021; Korotkova and Jakobsson, 2014; Korotkova and Lundberg, 2014). Physiologically, the

majority of ARA binds to membrane phospholipids, preventing the fatty acid from serving as oxidative substrates (Pérez et al, 2006). Thus, under resting conditions, eicosanoid production is low (Calder, 2020). However, in the presence of some stimuli, sufficient ARA is released by $PLA_2$ to drive significant increases in eicosanoid formation (Calder, 2020; Fierens and Kool, 2012). Hence, in this situation, $PLA_2$ activity was increased in the draining lymph nodes under the stimulation of immunization (Fierens and Kool, 2012), and free ARA can be converted to $PGI_2$ by a cascade of enzymatic reactions. Although ARA is administered systemically, the absence of sufficient physiological stimulation in individuals may explain the unchanged $PGI_2$ levels in plasma. This observation aligns with previous investigations, where larger doses and longer ARA supplementation resulted in no changes in eicosanoids derived from ARA in the plasma (Kakutani et al, 2011).

According to the routine immunization program for inactivated or subunit vaccines, multiple-shot administration is usually necessary to boost protective immunity against infectious diseases. Nonetheless, a salient challenge of routine vaccination is that a long interval is required for generating robust immune responses to prevent infection. This temporal interval, denoted as the window of vulnerability, potentially permits pathogenic invasion prior to the establishment of a protective immune response. For example, RABV, mostly transmitted by animal bites, may enter the peripheral nervous system through the neuromuscular junction and travel centripetally along the spinal cord to the brain to infect the whole central nervous system (CNS) (Lafon, 2005). Lethal infection may result from the failure of the Rabies vaccine to produce protective anti-RABV antibodies in a timely manner (Fooks et al, 2017). Therefore, accelerating the production of neutralizing antibodies is essential to promoting the protective efficacy of the rabies vaccine. In this study, we demonstrated that dietary administration of ARA enabled vaccinated animals to acquire humoral immune protection in the early time, meanwhile largely enhancing the production of anti-RABV antibodies after immunization with the rabies vaccine. Thus, this administrative strategy offered a way to shorten the window of vulnerability after vaccination and strengthen the vaccine's protective effects. These results exemplify the effectiveness of dietary ARA for boosting and accelerating humoral immune responses in the trial of immunization in humans. Given the function of ARA supplementation in promoting the GC responses, we speculate that this dietary supplementation strategy may also be applied to boost humoral immunity with other vaccines, particularly those against emerging infectious diseases. For example, the annual or semi-annual flu shots encounter a similar circumstance. As influenza viruses constantly evolve, each flu season is likely caused by a new strain that requires re-immunization and rapid production of antibodies to avoid infection (Dos Santos et al, 2016).

In summary, vaccines are pivotal in safeguarding public health by orchestrating the immune response and fostering the production of antibodies against specific pathogens. Here, we devise a safe strategy with dietary supplementation of a fatty acid, thereby expediting the activation of GC responses and the production of neutralizing antibodies after vaccination, which minimizes the window of vulnerability. This research holds the potential to yield more efficacious vaccines and ultimately better control infectious diseases worldwide.

# Methods

**Reagents and tools table**

| Reagent/resource | Reference or source | Identifier or catalog number |
|---|---|---|
| **Experimental models** | | |
| BALB/c | Beijing Vital River Laboratory Animal Technology Co., Ltd. | |
| BHK-21 cells (M. musculus) | ATCC | Stock No#CCL-10 |
| Cell lines N2a (M. musculus) | ATCC | Stock No#CCL-131 |
| **Recombinant DNA** | | |
| NA | | |
| **Antibodies** | | |
| Ghost Dye™ UV 450 | TONBO Biosciences | Cat #13-0868-T500 |
| PE/Cyanine5 anti-mouse CD19 | Biolegend | Cat #115510 |
| APC/Cyanine7 anti-mouse/human B220 | Biolegend | Cat #103224 |
| APC anti-mouse/human GL7 | Biolegend | Cat #144618 |
| FITC anti-mouse IgG1 Antibody | Biolegend | Cat #406606 |
| PerCP/Cyanine5.5 anti-mouse IgG1 | Biolegend | Cat #406612 |
| Pacific Blue™ anti-mouse CD3 | Biolegend | Cat #100214 |
| Brilliant Violet 510™ anti-mouse CD4 | Biolegend | Cat #100449 |
| Alexa Fluor® 594 anti-mouse CD8a | Biolegend | Cat #100758 |
| Alexa Fluor® 700 anti-mouse/human CD44 | Biolegend | Cat #103026 |
| Brilliant Violet 605™ anti-mouse CD185 | Biolegend | Cat #145513 |
| Brilliant Violet 421™ anti-mouse CD279 | Biolegend | Cat #135221 |
| FITC anti-mouse CD38 | Biolegend | Cat #165608 |
| PerCP/Cyanine5.5 anti-mouse IgD | Biolegend | Cat #405710 |
| PE/Cyanine5 anti-human CD19 | Biolegend | Cat #363042 |
| Alexa Fluor® 700 anti-human CD20 | Biolegend | Cat #302322 |
| APC anti-human CD138 | Biolegend | Cat #352308 |
| APC/Cyanine7 anti-human CD38 | Biolegend | Cat #356616 |
| PE anti-human IgD | Biolegend | Cat #348204 |
| Brilliant Violet 421™ anti-human CD27 | Biolegend | Cat #356418 |
| FITC anti-human CD3 | Biolegend | Cat #317306 |
| Brilliant Violet 605™ anti-human CD8 | Biolegend | Cat #344742 |
| OVA-PE | Bioss | Cat #bs-0283P-PE |

| Reagent/resource | Reference or source | Identifier or catalog number |
|---|---|---|
| OVA-Alexa Fluor™ 647 | Invitrogen | Cat #O34784 |
| AF488-conjugated goat anti-rabbit IgG | Servicebio | Cat #GB22303 |
| HRP-conjugated Goat anti-Rabbit IgG (H + L) | Abclonal | Cat #AS014 |
| HRP-anti-mouse IgG | Abclonal | Cat #AS003 |
| HRP-goat anti-mouse IgG | Proteintech | Cat #SA00001-1 |
| HRP-anti-human IgG | Proteintech | Cat #SA00001-17 |
| AID (L7E7) Mouse mAb | CST | Cat #4975S |
| CD86 Rabbit pAb | Abclonal | Cat #A16805 |
| F(ab')2 Fragment Goat Anti-Mouse IgM | Jackson | Cat #JAC-115-006-20 |
| Purified anti-mouse CD40 Antibody | Biolegend | Cat #102802 |
| Purified anti-CD3 Antibody | Biolegend | Cat #830301 |
| Purified anti-mouse CD28 Antibody | Biolegend | Cat #102102 |
| **Oligonucleotides and other sequence-based reagents** | | |
| RABV *N* gene forward primer | This study | AGGTGACAGCATTGCTTCTG |
| RABV *N* gene reverse primer | This study | GCTGCCTCAACACCTCAAC |
| **Chemicals, enzymes, and other reagents** | | |
| Arachidonic Acid (ARA) | Apexbio | Cat #C4223 |
| Eicosapentaenoic Acid (EPA) | Apexbio | Cat #B3464 |
| Linoleic Acid (LA) | Apexbio | Cat #C3108 |
| γ-Linolenic Acid (GLA) | Apexbio | Cat #C5518 |
| α-Linolenic Acid (ALA) | Apexbio | Cat #C3934 |
| Docosahexaenoic Acid (DHA) | Apexbio | Cat #C4188 |
| ( ± )14,15-Epoxyeicosatrienoic acid (14(15)-EET) | Apexbio | Cat #C4455 |
| ( ± )17-Hydroxyeicosatetraenoic acid (17-HETE) | Apexbio | Cat #C4710 |
| ( ± )18-Hydroxyeicosatetraenoic acid (18-HETE) | Apexbio | Cat #C4706 |
| 12(R)-Hydroxyeicosatetraenoic acid (12(R)-HETE) | Apexbio | Cat #C5362 |
| 8(R)-Hydroxyeicosatetraenoic acid (8(R)-HETE) | Apexbio | Cat #C5471 |
| ( ± )9-Hydroxyeicosatetraenoic acid ((±)9-HETE) | Apexbio | Cat #C5479 |
| 11(R)-Hydroxyeicosatetraenoic acid (11(R)-HETE) | Apexbio | Cat #C5486 |
| ( ± )5,6-Epoxyeicosatrienoic acid ((±)5(6)-EET) | Apexbio | Cat #C3284 |
| 15(S)-Hydroxyeicosatetraenoic acid (15(S)-HETE) | Apexbio | Cat #C4187 |
| 16(R)-Hydroxyeicosatetraenoic acid (16(R)-HETE) | Apexbio | Cat #C4673 |

| Reagent/resource | Reference or source | Identifier or catalog number |
|---|---|---|
| ( ± )8(9)-Epoxyeicosatrienoic acid (8(9)-EET) | Apexbio | Cat #C3437 |
| ( ± )11(12)-Epoxyeicosatrienoic acid (11(12)-EET) | Apexbio | Cat #C5501 |
| Prostaglandin D2 (PGD2) | Apexbio | Cat #C7200 |
| Prostaglandin E2 (PGE2) | Apexbio | Cat #B7005 |
| Leukotriene E4 (LTE4) | Aladdin | Cat #L274987 |
| Leukotriene D4 (LTD4) | Aladdin | Cat #L275075 |
| ( ± )5-Hydroxyeicosatetraenoic acid (5-HETE) | Aladdin | Cat #H336046 |
| 6-keto-Prostaglandin F1α (6-keto-PGF1α) | Aladdin | Cat #K336345 |
| 15-Deoxy-Δ-12,14-prostaglandin J2 (15d-PGJ2) | Aladdin | Cat #D275807 |
| Leukotriene B4 (LTB4) | MCE | Cat #HY-107608 |
| Thromboxane B2 (TxB2) | MCE | Cat #HY-113331 |
| 15-keto-Prostaglandin E2 (15-keto-PGE2) | GLPBIO | Cat #GC40605 |
| Prostaglandin F2α (PGF2α) | Macklin | Cat #P868175 |
| Leukotriene C4 (LTC4) | Macklin | Cat #L912678 |
| Anandamide (AEA) | Meryer | Cat #M62704 |
| Aluminum hydroxide gel | InvivoGen | Cat #vac-alu-250 |
| Ovalbumin | Sigma-Aldrich | Cat #9006-59-1 |
| Lipopolysaccharides | MCE | Cat #HY-D1056 |
| Beta-propiolactone | Sigma | 219126 |
| fetal bovine serum | Sigma | F0193 |
| Recombinant Mouse IL-4 Protein | Sino biological | Cat #51084-NAE |
| Recombinant Mouse IL-2 Protein | Sino biological | Cat #51061-NAE |
| CellTrace Violet dye | Invitrogen | Cat #C34557 |
| Pre-stained Protein Marker | Sangon Biotech | Cat #C610016 |
| Ovalbumin Peptide (323-339) | MCE | Cat #HY-P0286-1mg |
| Paraformaldehyde | Solarbio | Cat #P1110 |
| Beraprost | MCE | Cat #HY-13569A |
| H89 | MCE | Cat #HY-15979 |
| Forskolin | MCE | Cat #HY-15371 |
| MDL12330A | MCE | Cat #HY-103192 |
| 6-Bnz-cAMP sodium salt | MCE | Cat #HY-103322 |
| U-46619 | MCE | Cat #HY-108566 |
| MultiScreenHTS IP Filter Plate | Merckmillipore | Cat #MSIPS4510 |
| 96-well Clear Polystyrene Microplates | Corning | Cat #3690 |
| Osmotic Pumps | Alzet | Cat #2006 |
| Immunoprecipitation Kit with Protein A Magnetic Beads | Beyotime | Cat #P2175S |
| Solid-phase extraction cartridges | Waters | Cat #WAT054945 |
| TMB substrate for ELISpot | MabTech | Cat #3651-10 |

| Reagent/resource | Reference or source | Identifier or catalog number |
|---|---|---|
| 5X ELISA/ELISPOT DILUENT | Thermo Fisher | Cat #00-4202-56 |
| Methanol | TCI | Cat #M0628 |
| Dichloromethane | TCI | Cat #M0629 |
| Hexane | TCI | Cat #H0490 |
| Formic Acid | TCI | Cat #F0654 |
| **Software** | | |
| FlowJo v10 | https://www.flowjo.com/solutions/flowjo/ | |
| Prism v8 | https://www.graphpad.com/ | |
| **Other** | | |
| EasySep Mouse B Cell Isolation Kit | StemCell | Cat #19854 |
| EasySep Mouse CD4 T Cell Isolation Kit | StemCell | Cat #19852 |
| cAMP ELISA kit | Jianglai | Cat #JL13362-96T |
| Protein Kinase A Colorimetric Activity Kit | InvivoGen | Cat #EIAPKA |
| AID Elispot Reader-iSpot | AID | |
| Light Sheet Fluorescence Microscopy | Zeiss | |
| ID7000™ Spectral Cell Analyzer | SONY | |
| BD FACSymphony™ S6 SE Cell Sorter | BD Biosciences | |

## Methods and protocols

### Cells

Cell lines N2a (murine neuroblastoma N2a cells, ATCC®CCL-131) and BSR (a cloned cell line derived from BHK-21, ATCC®CCL-10) were cultured in Dulbecco's Modified Eagle Medium (DMEM, Gibco, USA) supplemented with 10% bovine serum (Gibco) at 37 °C with 5% $CO_2$.

### Viruses

The RABV vaccine strain SAD-L16 was obtained by reverse inheritance of the weakly virulent strain SAD-B19 (Gen-Bank:M31046.1) as previously described (Schnell et al, 1994). It contains two mutations in the G protein at amino acid positions 194 and 333 compared to the parent virus. The rabies challenge virus (CVS-11) for rabies virus neutralizing antibody (VNA) titer measurement was prepared and stored in prof. Zhao's lab. DRV-Mexico, a dog-derived RABV wild-type strain, was isolated from a human patient and propagated in suckling mouse brains (Yu et al, 2014), and was used for mouse challenge experiments.

### Mice

Female BALB/c mice (6–8 weeks old) were purchased from Beijing Vital River Laboratory Animal Technology Co., Ltd. and kept in a specific pathogen-free animal facility at the Laboratory Animal Resources Center, Tsinghua University. All animal experiments were approved by the Institutional Animal Care and Use Committee and conducted in accordance with governmental and Tsinghua guidelines for animal welfare.

### Mouse immunization

Mice were vaccinated with OVA (20 µg, Sigma) suspended in PBS containing alum (500 µg, InvivoGen) with a final injection volume of 100 µL. For RABV immunization, the RABV vaccine strain SAD-L16 was inactivated with 0.025% (v/v) beta-propiolactone (BPL) (Sigma-Aldrich, Darmstadt) at 4 °C for 24 h. The residual BPL was hydrolyzed in the water bath at 37 °C for 2 h, and mice were immunized with 100 µL of a solution containing $10^7$ Focus-forming unit (FFU) of inactivated rabies vaccine. The vaccine doses used were referenced to the model as previously described (Chen et al, 2019; Wang et al, 2021; Zhang et al, 2020). All injections were performed intramuscularly (i.m.). The mice were orally or subcutaneously administered polyunsaturated fatty acids for 13 days. Mice were immunized with the antigens above on day 3. As described below, the samples were collected at the appointed time for various assays.

### In vitro stimulation of B and T cells

According to the manufacturer's instructions, mouse splenic B cells were isolated from splenocytes using the EasySep B Cell Isolation Kit (STEMCELL Technologies, Canada). FO B cells were further enriched from the isolated B cells with BD FACSymphony™ S6 SE Cell Sorter and diluted in complete Roswell Park Memorial Institute (RPMI) 1640 (containing 10% heat-inactivated Fetal Bovine Serum (FBS), 55 µM β-mercaptoethanol, 10 mM Hepes, 2 mM glutamine, and 50 IU penicillin/streptomycin). The B cells were seeded at a density of $5 \times 10^5$ cells in 96-well round-bottom plates and were stimulated with LPS (25 mg/mL) plus IL-4 (10 ng/mL), or with anti-CD40 (1 mg/mL) plus IL-4 (10 ng/mL) and anti-IgM (1 µg/mL). Mouse splenic CD4 T cells were purified by negative selection with the EasySep Mouse T cell Isolation Kit (StemCell Technologies, Canada), and the T cells were plated at a density of $5 \times 10^5$ cells in 96-well round-bottom plates and stimulated with anti-CD3 (1 µg/mL), anti-CD28 (2 µg/mL), and IL-2 (5 ng/mL). B cells and T Cells were labeled with the 5 µM CellTrace Violet dye (Thermo Fisher Scientific). On day 3 or 4, the proliferation of the stimulated B cells and T cells or the IgG1 class switching of the stimulated B cells were assessed by flow cytometric analysis on ID7000 Spectral Cell Analyzer. For T cells restimulation in vitro, splenocytes were obtained from mice 9 and 14 days after immunization, seeded into 96-well round-bottom plates at a density of $5 \times 10^5$ cells and restimulated with $OVA_{323-339}$ peptides (1 µM, MCE) for 18 h. The intracellular cytokines from T cells were detected by flow cytometry.

### Liquid chromatography-tandem mass spectrometry (LC-MS/MS)

For PUFAs analysis: Fatty acid extraction was conducted following previously published methodology (23671091). Briefly, 400 µL of a 2:1 (v/v) dichloromethane/methanol solution was combined with 100 µL of serum and subjected to vigorous vortex-mixing for 30 s. The mixture was subsequently incubated for 5 min; this extraction procedure was repeated twice to ensure complete extraction. Following extraction, the mixture was centrifuged at room temperature (3000 rpm for 20 min). After phase separation, an equivalent volume of the organic phase (lower layer) was carefully transferred to a new 1.5 mL EP tube. The extract was evaporated to dryness under a gentle

stream of $N_2$. Add 1 mL of a 90:10 (v/v) methanol/KOH mixture containing 0.3 M KOH to resuspend the sample, incubated at 80 °C for 1 h to saponify fatty acids, acidified with 0.1 mL of formic acid, extracted twice with 1 mL of hexane, dried under $N_2$. The resultant residue was reconstituted in 80 μL of isopropanol (IPA)/methanol (MeOH) (v:v = 9:1). Lipid analysis was performed using an UPLC system coupled to an Orbitrap Exploris 240 mass spectrometer (Thermo Fisher, CA) equipped with a heated electrospray ionization (HESI) probe. Lipid extracts were chromatographically separated on a CORTECS C18 column (100 × 2.1 mm, 2.7 μm; Waters, USA). A binary mobile phase system was employed: mobile phase A consisted of acetonitrile:water (60:40) with 10 mM ammonium acetate, and mobile phase B comprised isopropanol:acetonitrile (90:10). Separation utilized a 10-minute linear gradient at a flow rate of 250 μL/min as follows: 0 min, 5% B; 2.5 min, 5% B; 3.5 min, 30% B; 5.5 min, 98% B; 8 min, 98% B; 8.1 min, 5% B; 10 min, 5% B. The column chamber and sample tray temperatures were maintained at 30 °C and 10 °C, respectively.

Data with mass ranges of $m/z$ 150–600 was acquired at negative ion mode. The full scan was collected with a resolution of 60,000. The source parameters are as follows: spray voltage: 3000 v; capillary temperature: 320oC; heater temperature: 300oC; sheath gas flow rate: 35 Arb; auxiliary gas flow rate: 10 Arb.

For ARA-metabolized eicosanoids analysis: Tissues were homogenized with 500 μL of 80% methanol (containing 2% formic acid and 0.01 mol/L butylated hydroxytoluene) and mixed on a vortexer for 5 min. After centrifugation (12,000× $g$ for 10 min at 4 °C), the supernatant was loaded into Solid-phase extraction cartridges (Sep-Pak, Waters, Milford, MA, USA), which were conditioned and equilibrated according to the manufacturer's instructions. Solid-phase extraction cartridges were washed with water and hexane successively, methyl formate was added for elution, and the obtained eluent solution was dried under nitrogen protection at room temperature (RT). Targeted lipidomics of the extract was analyzed by the 6500plus QTrap mass spectrometer (AB SCIEX, USA) coupled with ACQUITY UPLC H-Class system (Waters, USA). An ACQUITY Premier BEH C18 column (100 × 2.1 mm, 1.7 μm, Waters) was utilized with 0.1% acetic acid in $H_2O$ and Acetonitrile/Isopropanol, 90/10 (v/v) as mobile phase A and B, respectively. In this experiment, we used a 10-min gradient from 30 to 95% mobile B. Flow rate was 0.6 mL/min. Positive-negative ion switching mode was performed for data acquisition in a multiple reaction monitor (MRM). The resolution for Q1 and Q3 of Si Quadrants are both units. The source voltage was 5000 V for positive and −4500 V for negative ion mode. The ion transitions were optimized using chemical standards. The nebulizer gas (Gas1), heater gas (Gas2), and curtain gas were set at 50, 55, and 35 psi, respectively. The optimal probe temperature was determined to be 525 °C. The SCIEX OS 1.6 software was applied for metabolite identification and peak integration.

## cAMP and PKA kinase assays

FO B cells were incubated with Beraprost (500 nM) for 30 min, followed by washing twice with ice-cold PBS and lysed, and the cellular cAMP levels and PKA activity were detected by ELISA according to the manufacturer's instructions: cAMP ELISA kit (Jianglai, Shanghai, China); PKA activity kit (ThermoFisher Scientific).

## Immunofluorescence stain and microscopy

For GC analysis, inguinal lymph nodes from immunized mice on day 10 were collected into PBS and fixed with 4% paraformaldehyde for 1 h at 4 °C, followed by a five-time wash in PBS. Thereafter, lymph nodes were placed sequentially in 10% and 30% sucrose overnight at 4 °C and snap-frozen in cryomolds using Optimal Cutting Temperature compound OCT in a dry ice-cooled bath. For immunostaining, 30-μM tissue sections were cut and air-dried for 1 h before rehydration in PBS with 1% BSA for 10 min. Slides were then washed three times in PBST and blocked for 1 h at RT with 3% bovine serum albumin and subsequently stained with antibodies against CD3, GL7, and IgD in PBS with 0.1% BSA and 0.1% $NaN_3$ overnight at 4 °C. For analysis of viral load in the brain, the mice brains collected at day 12 post-infection were flash-frozen and sectioned into 30 μm slices as above. RABV was stained with polyclonal antibody against RABV-P (prepared in lab) and AF488-conjugated goat anti-rabbit IgG (Servicebio, GB25303, 1:500), and the nucleus was stained with 4',6-diamidino-2-phenylindole (DAPI). After washing in PBS, slides were mounted using ProLong™ Diamond Antifade (Invitrogen) and imaged using a Zeiss LSM710 confocal microscope. Tile scan images were taken using a 10X magnification lens with 10% overlap. Images were processed and analyzed using Zeiss Blue 3.1 (Zeiss) and Volocity 6.3 (Perkin Elmer) software.

## Western blots

Stimulated FO B cells were lysed using a lysis buffer containing 1× protease inhibitors. The extracted proteins were separated by 12% sodium dodecyl-sulfate polyacrylamide gel electrophoresis (SDS-PAGE) and then transferred onto the polyvinylidene fluoride membrane (PVDF, Bio-Rad). Next, the membrane was blocked with 5% BSA for 1 h at RT and then incubated with primary antibody overnight at 4 °C, followed by incubation with horseradish peroxidase (HRP)-conjugated goat anti-rabbit (AS014, ABclonal, 1:5000) or anti-mouse IgG (AS003, ABclonal, 1:5000) for 1 h at RT. The target bands were detected by incubating the membrane with a chemiluminescent substrate.

## Immunoprecipitation (IP) assays

Stimulated FO B cells were lysed in 500 μL IP lysis buffer. First, 50 μL lysates were transferred to a new tube for input assay, and the remaining lysates were incubated with anti-AID antibodies (4975S, CST, 1:1000) overnight at 4 °C, followed by incubation with protein G magnetic beads (P2177S, Beyotime) at 4 °C for 2 h. After washing with IP lysis buffer, the samples were boiled and subjected to western blots analysis.

## ELISA measurement of antibody titer

High-binding flat-bottom 96-well plates (3690, Corning Life Sciences, NY, USA) were coated with 5 μg/mL OVA or purified RABV (SAD-L16) virion in ELISA coating buffer under 4 °C overnight. Plates were then washed three times with phosphate-buffered saline (PBS)-Tween (PBST) (0.5% Tween 80) and blocked for 2 h at 37 °C with PBS containing 5% skim milk. After washing with PBST, serially diluted serum was added to the wells and incubated at 37 °C for 2 h. Plates were then washed and

incubated for 45 min at 37 °C with HRP-conjugated goat anti-mouse IgG (SA0001-1, Proteintech, 1:5000) or anti-human IgG (SA00001-17, Proteintech, 1:2000) antibodies. The plates were stained with tetra-methylbenzidine substrate (Biotime Biotechnology, Shanghai, China) in the dark for 10 min, and reactions were stopped with 2 M sulfuric acid. Finally, the optical density was measured at 450 nm using a SpectraMax 190 spectrophotometer (Molecular Devices, CA, USA).

## ELISpot assay

ELISpot plates (Millipore, Burlington, MA, USA) were coated with 100 μL of 5 μg/mL OVA or purified RABV overnight at 4 °C, followed by washing with PBS and blocking with RPMI 1640 containing 10% heat-inactivated FBS for 2 h in cell culture incubator. Single cell suspensions of lymphocytes from lymph nodes of mice or PBMCs of volunteers were added to the plates at different dilutions and incubated for 24 h, then washed five times with PBST, followed by incubation with HRP-goat anti-mouse IgG (1:5000) or anti-human IgG (1:2000) antibodies diluted in culture media for 2 h at RT. AEC substrate (3′ amino-9-ethylcarbazole; BD Bioscience) was added after washing to detect the spots. Images were acquired in an ELISpot/FluoroSpot reader system (MultiSpot Reader Spectrum, AID, Strassberg, Germany), and scored spots were counted.

## Flow cytometry and cell sorting

Mononuclear cells were harvested from the spleen and inguinal lymph nodes by mashing through a 70-μm cell strainer with a syringe plunger. Splenocytes were further incubated in 1 mL red blood cell lysis buffer (155 mM $NH_4Cl$, 10 mM $KHCO_3$, 0.1 mM EDTA) for 5 min at 4 °C to lyse before filtering through 40-μm cell strainer and resuspended in MACS buffer (2% heat-inactivated FCS in PBS with 1 mM EDTA). The obtained leukocytes were incubated for 10 min on ice with anti-CD16/32 Fc-block (1:200) and then stained with Ghost Dye™ UV 450 (TONBO Biosciences, Cat. No# 13-0868-T500) (1:500) for dead cells and then stained with the fluorochrome-coupled antibodies on ice for 30 min in MACS buffer. The stained cells were assessed on ID7000 Spectral Cell Analyzer. The gating strategies are illustrated in Appendix Fig. S3. All antibodies used in flow cytometry were purchased from BioLegend unless otherwise indicated. The following antibodies were used: PE/Cyanine5 anti-mouse CD19 (1:200), APC/Cyanine7 anti-mouse/human CD45R/B220 (1:200), APC anti-mouse/human GL7 (1:200), FITC anti-mouse IgG1 Antibody (1:300), PerCP/Cyanine5.5 anti-mouse IgG1 (1:300), Pacific Blue™ anti-mouse CD3 (1:300), Brilliant Violet 510™ anti-mouse CD4 (1:200), Alexa Fluor® 594 anti-mouse CD8a (1:300), Alexa Fluor® 700 anti-mouse/human CD44 (1:300), Brilliant Violet 605™ anti-mouse CD185 (CXCR5) (1:100), Brilliant Violet 421™ anti-mouse CD279 (PD-1) (1:100), FITC anti-mouse CD38 (1:300), PerCP/Cyanine5.5 anti-mouse IgD (1:100), PE/Cyanine5 anti-human CD19 (1:100), Alexa Fluor® 700 anti-human CD20 (1:200), APC anti-human CD138 (Syndecan-1) (1:100), APC/Cyanine7 anti-human CD38 (1:100), PE anti-human IgD (1:100), Brilliant Violet 421™ anti-human CD27 (1:100), FITC anti-human CD3 (1:100), and Brilliant Violet 605™ anti-human CD8(1:100). Direct detection of antigen-binding B cells was performed using OVA-PE (1:200) and OVA-Alexa Fluor™ 647 (1:200),or G-Brilliant Violet 421(1:100), labeling of single-cell suspensions. For sorting FO B cells, B cells were enriched using

EasySep B Cell Isolation Kit (StemCell) from mice splenocytes, and subsequently, cells (PE anti-mouse CD23[hi], APC anti-mouse CD1d[low]) (1:100) sorted with BD FACSymphony™ S6 SE Cell Sorter.

## Array analysis for cytokines

Serum was obtained from mice immunized with OVA on day 7 and day 14, respectively. Cytokines in serum were analyzed using a mouse cytokine array kit (R&D Systems; QAH-TH17-1, RayBiotech, Norcross, GA) according to the manufacturer's specification instructions. An Axon scanner 4000B with GenePix software collected fluorescence signals.

## Quantitative real-time PCR

The mRNA level of RABV *N* gene in mouse brains at day 8 and day 12 post-infection was evaluated by quantitative real-time PCR (qPCR). The total RNA of the whole brain, cortex, cerebellum, brainstem, or olfactory bulb was isolated using TRIzol reagent (Invitrogen, Karlsruhe, Germany). RNA was quantified using a NanoDrop and Agilent 2100 bioanalyzer (Thermo Fisher Scientific, MA, USA) and converted to cDNA by reverse transcription using FSQ-201 ReverTra Ace (TOYOBO, Osaka, Japan). qPCR was performed using SYBR green Supermix (Bio-Rad; 172-5124) on an Applied Biosystems 7300 real-time PCR system (Applied Biosystems, CA, USA) with the primes RABV-N-F (5'-AGGTGACAG-CATTGCTTCTG-3') and RABV-N-R (5'-GCTGCCTCAACACC TCAAC-3').

## Virus titration in mouse brain

The mice brains were collected at day 8 and day 12 post-infection. Mouse brains were homogenized (20% w/v) with phosphate-buffered saline (PBS) containing 2% fetal bovine serum (FBS) and centrifuged at 12,000 r/min for 10 min at 4 °C. The supernatant was collected, and the viral titers were determined with the focus assay. Briefly, the supernatant was diluted fivefold and placed in plates in quadruplicate, then N2a cells were added and incubated at 37 °C with 5% $CO_2$ for 24 h. The plates were fixed with 80% cold acetone at −20 °C for 30 min, and incubated with polyclonal antibody against RABV-P (prepared in lab) and AF488-conjugated goat anti-rabbit IgG (Servicebio, Wuhan, China) at 37 °C for 45 min. Positive foci were calculated under a fluorescence microscope, and virus titers were calculated and expressed as focus-forming units per ml (FFU/mL).

## RABV neutralizing antibody (VNA) titers measurement

Rabies VNA titers were measured by the fluorescent-antibody virus neutralization (FAVN) assay. Briefly, the serum samples were separated and heat-inactivated at 56 °C for 30 min. Serial dilutions of both the test serum and standard serum were prepared in 96-well microplates. Each sample was added to four adjacent wells. Subsequently, the suspension of the rabies challenge virus (CVS-11) was added into each well. The plates were incubated at 37 °C for 1 h and then cultured with $2 \times 10^4$ BSR cells each well at 37 °C for 60 h. The samples were fixed with pre-cooled acetone (80%, w/w) for 30 min and stained with FITC-RABV N antibodies (800-092, Fujirebio Diagnostics, 1:500). The fluorescence values were observed by Olympus IX51 fluorescence microscope (Olympus,

Japan), and compared with the standard serum obtained from the National Institute for Biological Standards and Control (Herts, United Kingdom). The antibody titers were quantified in international units per milliliter.

## Passive transfer model in mice

BALB/c mice were treated daily with individual PUFAs or PBS control for 3 days; the animals were then administered a single-dose ($1 \times 10^7$ FFU in 100 μL) inactivated rabies vaccine and continued the PUFA/PBS treatment for 7 days ($n = 16$). At 3 weeks post-vaccination, the serum was collected and heat-inactivated at 56 °C for 30 min. Six-week-old BALB/c mice were intraperitoneally (i.p.) administered with 200 μL serum. One day later, mice were intramuscularly (i.m.) challenged with 100 $LD_{50}$ of RABV. The clinical scores and survival were monitored daily for 21 days ($n = 8$).

## Randomized, triple-blinded, placebo-controlled parallel arm intervention trial

### Study design

All procedures involving human participants were approved by the Institutional Review Board of Tsinghua University (Project No: 20220081). The clinical trial was registered on ClinicalTrials.gov. (NCT05987384). Before commencement of the study, the experimental protocol was explained to the participants, and informed consent was obtained from all human subjects. Our human study was conducted in accordance with the WMA Declaration of Helsinki and the Department of Health and Human Services Belmont Report. A randomized, triple-blinded, placebo-controlled parallel arm intervention trial was conducted, in which participants were randomly assigned to one of 3 groups to receive nutritional supplementation or a placebo for 17 days, along with vaccination for rabies.

This study included three periods: washout (day −6 to −4), supplementation (day −3/0 to 13), and post-supplementation (days 14–21). In the washout period, participants in all three groups were required to follow an ARA-restricted diet to minimize the individual variation of ARA intake at a baseline. Detailed dietary instructions are provided in Appendix Table S8. The supplementation period had no further dietary restrictions, and the participants were asked to keep their habitual diets. The control group received 6 capsules containing sunflower seed oil daily as a placebo; one intervention group received six capsules containing 512.4 mg of ARA daily (Pre-ARA group) from day −3 to 13; the other intervention group received the same supplementation as the Pre-ARA group (ARA group) from day 0 to 13. However, they received a placebo in the first 3 days of supplementation (day −3 to −1) to prevent unblinding. The ARA/placebo supplements used in this study were produced by CABIO Biotech (Wuhan) Co., Ltd. (Hubei, China), which passed the composition determination and quality inspection. The components of the capsule in each group are shown in Appendix Table S9. The intervention dose of ARA was calculated according to the results from animal experiments and human equivalent dose formula (Formula 1 and 2), where Km stands for converting factor (US Food and Drug Administration, 2005). All participants were instructed to take three capsules after breakfast and dinner. Compliance was assessed by logging attendance after daily administration and by taking pictures of empty bottles at the

end of the intervention. The RABV vaccine was injected on day 0 and 14, respectively, using Rabies Vaccine (Vero Cell) for Human Use (0.5 mL, 2.5 IU) provided by Liaoning Chengda Biotechnology Co., Ltd. The titers of anti-RABV IgG and VNA were determined at 7, 10, and 14 days after the first-shot vaccination and 7 days after the booster vaccination (day 21) at Huazhong Agricultural University.

$$\text{human dose(mg/kg)} = \text{animal dose(mg/kg)} \times \text{animal Km/human Km} \tag{1}$$

$$Km(kg/m^2) = \text{body weight(kg)}/\text{body surface area}^2(m^2) \tag{2}$$

### Participants

**Sample size:** According to the previous animal model, the mean (SD) of specific antibodies in the experimental and control group was 30810.77 (SD: 9444.20) and 11567.88 (SD: 6637.28), respectively. With reference to the above results, it was determined that the expected difference between the outcome indicators of the experimental and control groups in this study was 19242.89, and the SD was 12685.64 according to the SD combined formula (Formula 3) (Chandler et al, 2019). Taking the test level α = 0.05 and the degree of certainty 1-β = 0.9, $N_e$:$N_c$ = 1:1, and a cut-off level of superiority (δ) = 3000, the minimum sample size for each group was calculated as 11 according to the sample size calculation formula for superiority randomized controlled trials (Formula 4). Setting the expected loss of follow-up rate at 20%, 14 people were needed to be recruited in each group. Finally, 45 healthy participants aged 18–45 years were recruited by an invitation poster, and randomly assigned to one of the three groups.

$$SD = \sqrt{\frac{(N_1-1)SD_1{}^2 + (N_2-1)SD_2{}^2 + \frac{N_1 N_2}{N_1+N_2}(M_1{}^2 + M_2{}^2 - 2M_1 M_2)}{N_1 + N_2 - 1}} \tag{3}$$

$$N_e = \frac{(Z_{1-\alpha} + Z_{1-\beta})^2 \sigma^2 (1 + \frac{1}{k})}{(d - \delta)^2} \tag{4}$$

$$N_e = k \cdot N_c \tag{5}$$

## Inclusion and exclusion criteria

Inclusion criteria for subjects were as follows: (1) 18–45 years old; (2) body mass index (BMI) between 18.5 and 25.0 $kg/m^2$; (3) never had a history of rabies vaccination injection. The participants meeting the following criteria were excluded: (1) had severe disorders of abnormal lipid metabolism; (2) used lipid-lowering drugs, weight control drugs, and insulin drugs in the past three months; (3) received other vaccines in the past 3 months; (4) used probiotics or prebiotics in the past 3 months; (5) used steroids, immunosuppressants, and other hormonal drugs in the past year; (6) had any immunodeficiency diseases; (7) had a history of severe vaccine allergies; (8) disorders of liver and kidney metabolism; (9) had fever, cold, severe diarrhea, and other diseases in the past month; (10) smoking in the last year. In addition, drop-out criteria

were as follows: (1) use of other nutritional supplements during the intervention; (2) did not consume the ARA supplements provided by this study more than three times; (3) developed a severe illness; (4) experienced severe vaccine allergy; (5) could not complete the trial; (6) voluntary withdrawal.

### Randomizing and concealment

Forty-five participants met the inclusive criteria. The participants were assigned in a 1:1:1 ratio to one of the three masked groups based on the random number method by R version 4.3.1 (R Development Core Team, Vienna, Australia). The above operation and randomization codes for all the participants were held by two different investigators not involved in this study, and the information regarding the assignments was masked to researchers and participants until all data were collected and analyzed.

### Basic information and dietary assessment

Demographic characteristics and lifestyles were assessed by a designed electronic questionnaire at baseline, including age (years), sex (male/female), monthly income (Chinese yuan), weight (kg), height (m), physical activity (MET-min/week), sleep time (h/day), alcohol consumption, and health situation by self-report. Three consecutive 24-hour dietary surveys were completed by well-trained investigators on days −6 to −4, days 5–7, and days 11–13. The Sixth Revised Standard Edition of China Food Composition Table was used as a reference for calculating energy and macronutrient intakes in the three groups (Yang et al, 2018).

### Safety assessment

To evaluate the intervention safety, serum lipid (including triglycerides, total cholesterol, high-density lipoprotein cholesterol, and low-density lipoprotein cholesterol), blood routine (platelet count), blood coagulation profiles (including prothrombin time, activated partial thromboplastin time, thromboplastin time, prothrombin time percentage activity, international normalized ratio of prothrombin time, and fibrinogen), and inflammatory index (C-reactive protein) were examined before and after ARA intervention (blood samples were collected at the morning of −3 day and 14). In addition, anxiety level and depression were evaluated using General Anxiety Disorder-7 items and Patient Health Questionaire-9 items, respectively. Adverse reactions and events were also recorded and immediately reported to researchers.

## Statistical analysis

All the statistical analyses were performed by R 4.3.1 (R Development Core Team, Vienna, Australia). The $P$ value of <0.05 (two-tailed) was considered statistically significant.

Mean (SD) and median ($Q_1$, $Q_3$) were used to describe continuous data with and without normal distributions, respectively for population characteristics (demographic characteristics, lifestyles, and nutrient intakes) and antibody avidity over time (rabies virus neutralizing and specific antibodies). One-way analysis of variance (ANOVA) was performed to compare the statistical differences in population characteristics and antibody avidity among the three groups for continuous data with normal distribution and homoscedasticity, and the continuous data that did not meet the above

### The paper explained

#### Problem

Vaccines offer protection to individuals against pathogen infections, thereby underscoring their pivotal role in maintaining public health and global health security. According to the routine immunization program for inactivated or subunit vaccines, multiple-shot administration is usually necessary to boost protective humoral immunity against infectious diseases. Nonetheless, a salient challenge of routine vaccination is that a long interval is required to generate robust immune responses to prevent infection. This temporal interval, denoted as the window of vulnerability, potentially permits pathogenic invasion before establishing a protective immune response. This long-time window of vulnerability is generally acceptable with routine immunizations, but it becomes problematic in an emergency such as the COVID-19 pandemic when rapid induction of protective immunity is essential.

#### Results

This study demonstrated that supplemented ARA accumulates in lymph nodes, where it is metabolized into various eicosanoids. One key ARA metabolite, prostaglandin $I_2$ ($PGI_2$), acts via the cyclic adenosine monophosphate (cAMP)-protein kinase A (PKA) axis to upregulate expression of the costimulatory molecule CD86 and activate activation-induced cytidine deaminase (AID) in B cells, ultimately promoting the humoral immune response. In mice immunized with rabies vaccine, oral ARA administration induced a robust humoral immune response that protected against lethal rabies virus (RABV) infection. Furthermore, in human volunteers, oral ARA supplementation accelerated the development of neutralizing antibodies, reaching levels sufficient for protection against RABV as early as 1 week after primary immunization.

#### Impact

Vaccines are pivotal in safeguarding public health by orchestrating the immune response and fostering the production of antibodies against specific pathogens. Here, we devise a safe strategy with dietary supplementation of a fatty acid, thereby expediting the activation of B cell maturation and the production of neutralizing antibodies after vaccination. Distinct from alternative vaccine optimization approaches, ARA functions as an orally administered immunostimulant, offering a mechanism to mitigate the susceptible interval following immunization and enhance overall vaccine protective efficacy. This research suggests a potential nutritional strategy for augmenting vaccine effectiveness.

assumptions were tested by the Kruskal–Wallis test. For categorical data, the Chi-square test was used.

## Data availability

The datasets produced in this study are available in the following databases: Flow cytometry dataset: Figshare dataset at https://doi.org/10.6084/m9.figshare.29502476.v1.

The source data of this paper are collected in the following database record: biostudies:S-SCDT-10_1038-S44321-025-00310-7.

## Peer review information

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

## Acknowledgements

We thank the Core Facility of Center of Biomedical Analysis and the Metabolomics Facility Center of Metabolomics and Lipidomics in National Protein Science Technology Center of Tsinghua University for flow cytometry and LC-MS/MS experiments, respectively. This study was supported by grants from the National Natural Science Foundation of China (32188101), the Shenzhen Medical Research Fund (B2404002), the Yunnan Major Scientific and Technological Projects (202502AU100001), the National Natural Science Foundation of China (82271872, 82341046), Shenzhen San-Ming Project for Prevention and Research on Vector-borne Diseases (SZSM202211023), Yunnan Provincial Science and Technology Project at Southwest United Graduate School (202302AO370010), Fundamental Research Fund for the Central Universities (2662025DKPY009). This study was supported by the New Cornerstone Science Foundation through the New Cornerstone Investigator Program and the XPLORER PRIZE.

## Author contributions

**Shengyong Feng**: Data curation; Formal analysis; Investigation; Visualization; Methodology; Writing—original draft. **Enhao Ma**: Data curation; Formal analysis; Investigation; Methodology; Writing—original draft. **Xiaona Na**: Data curation; Formal analysis; Visualization; Methodology. **Zongmei Wang**: Data curation; Investigation; Visualization; Writing—original draft. **Wanbo Tai**: Resources; Investigation. **Xinhui Bao**: Investigation; Methodology. **Mao Wang**: Software; Methodology. **Han Chang**: Investigation; Methodology. **Baolei Wu**: Software; Methodology. **Miaoxi Liu**: Formal analysis; Investigation. **Juzhen Li**: Investigation. **Huicheng Shi**: Software; Formal analysis. **Celi Yang**: Investigation. **Menglu Xi**: Investigation. **Haibing Yang**: Investigation. **Yuhan Li**: Investigation. **Yibin Zhu**: Resources; Methodology. **Penghua Wang**: Methodology; Writing—review and editing. **Ling Zhao**: Resources; Supervision; Methodology. **Ai Zhao**: Conceptualization; Supervision; Methodology; Project administration. **Gong Cheng**: Conceptualization; Supervision; Funding acquisition; Project administration; Writing—review and editing.

Source data underlying figure panels in this paper may have individual authorship assigned. Where available, figure panel/source data authorship is listed in the following database record: biostudies:S-SCDT-10_1038-S44321-025-00310-7.

## Disclosure and competing interests statement

The authors declare no competing interests.

# Expanded View Figures

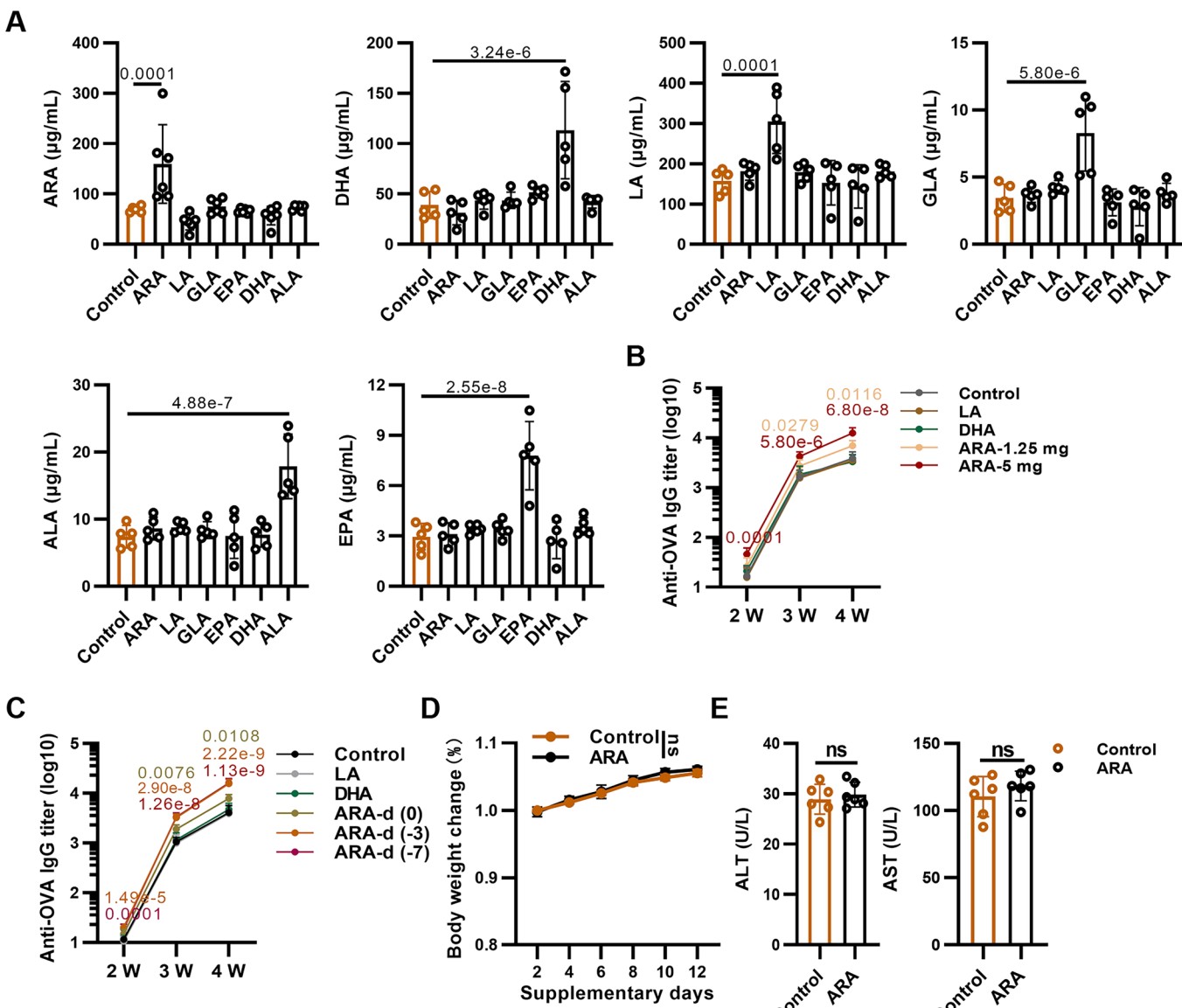

**Figure EV1. Exploring optimal settings for oral supplementation of ARA.**

(A) Fatty acid composition in murine plasma following supplementation with various PUFAs ($n = 6$ or $n = 5$). (B) OVA-specific IgG titers in mice orally administered serial doses of ARA at 2, 3, and 4 weeks post-immunization with OVA. DHA and LA were supplemented as unrelated controls ($n = 6$). (C) OVA-specific IgG titers in mice subjected to varying durations of ARA administration prior to immunization. DHA and LA were supplemented as unrelated controls ($n = 6$). (D) Changes in body weight of mice during ARA supplementation. (E) The levels of alanine aminotransferase (ALT) and aspartate aminotransferase (AST) of the mice supplemented with ARA under the optimal oral supplementation settings ($n = 6$). Data are representative of two independent experiments. Data are shown as mean ± SEM and each point represents an individual mouse. Significance was calculated by one-way ANOVA with Tukey's multiple comparisons test (A–C) and unpaired two-tailed $t$ test (D, E); ns, no statistical significance.

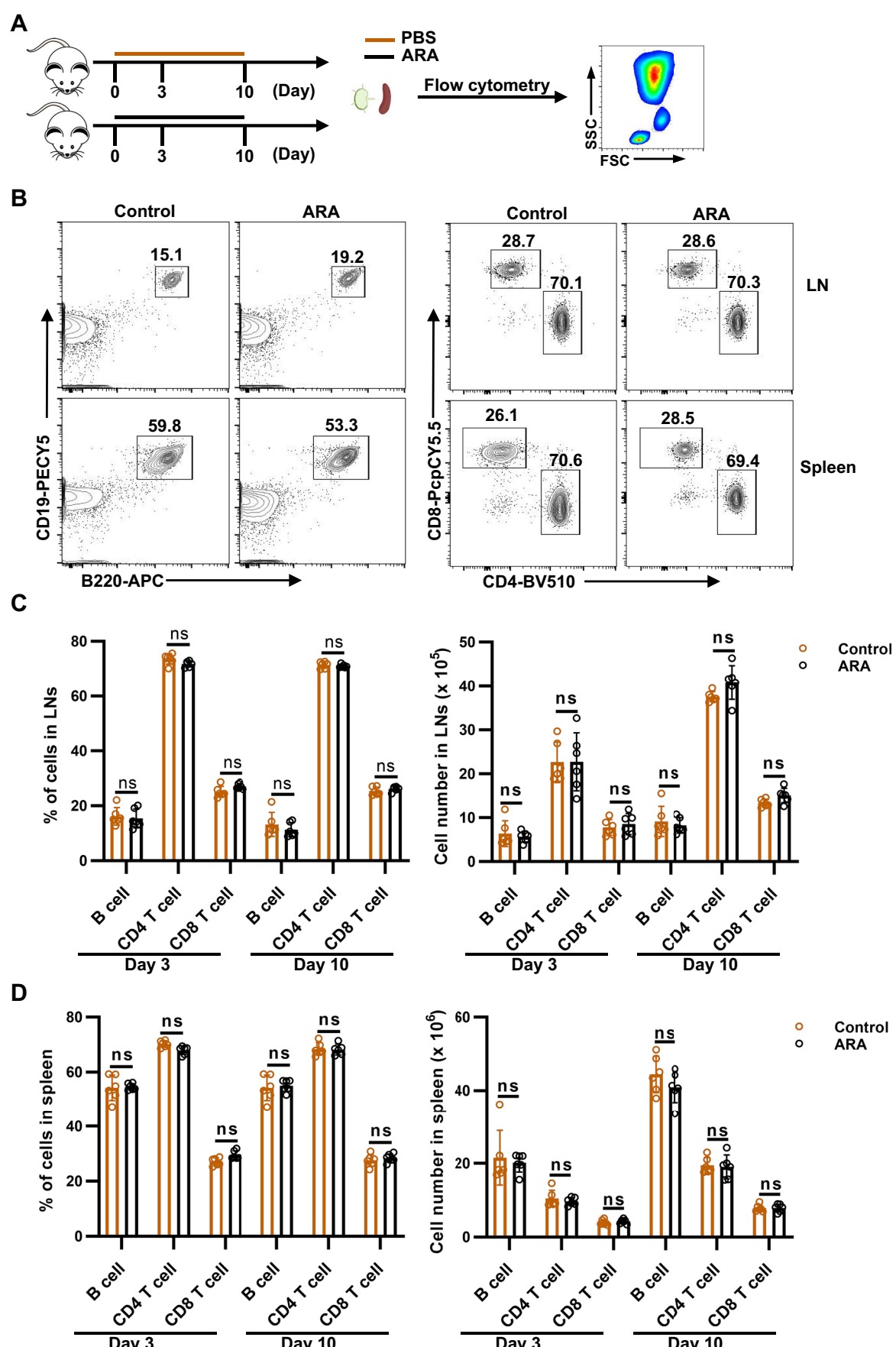

◀ **Figure EV2. The immune homeostasis was not influenced by daily ARA diet in mice spleen and lymph nodes.**

(A) Schematic diagram of the study design. Mice supplemented with ARA for 10 days, and lymph nodes and spleen were collected on day 3 and day 10 for flow cytometry analysis, respectively. Mice supplemented with PBS as control. (B) Representative flow cytometry plots of lymph nodes (LNs) and spleen from mice supplemented with ARA for 3 days and 10 days to identify total B cells (B220$^+$CD19$^+$), CD4 T cells (CD3$^+$ CD4$^+$) and CD8$^+$ T cells (CD3$^+$ CD8$^+$). (C, D) Percentages and absolute cell counts of B cells, CD4 T cells and CD8 T in LNs (C) and spleen (D) from mice supplemented with ARA for 3 days and 10 days ($n = 6$). Data are representative of two independent experiments. Data are shown as mean ± SEM and each point represents an individual mouse. Significance was calculated by unpaired two-tailed $t$ test; ns, no statistical significance.

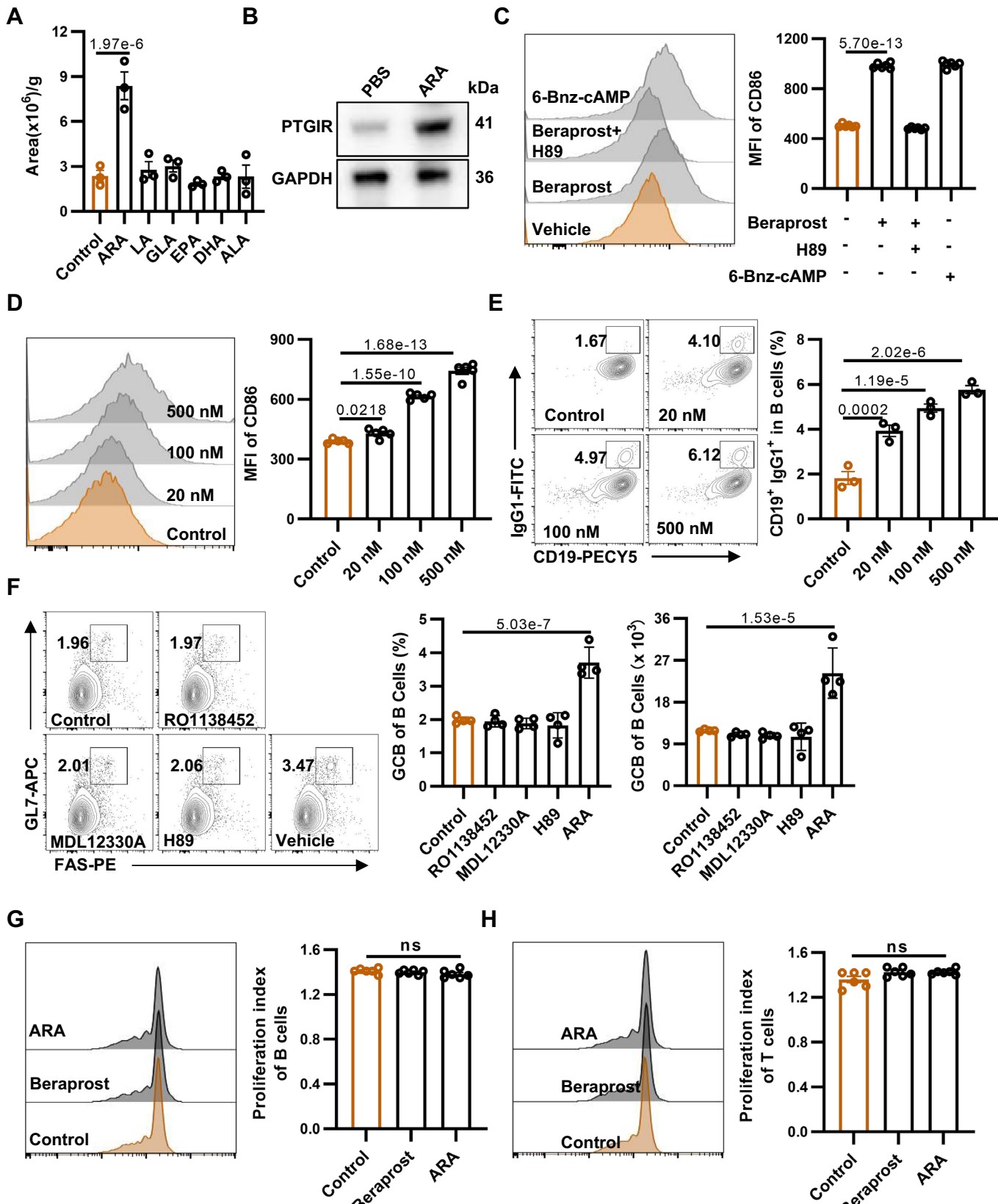

◀

**Figure EV3. PGI₂ derived from ARA promotes CD86 expression and enhances the activity of AID in FO B cells.**

(A) Quantification of PGI$_2$ from inguinal lymph nodes of mice supplemented with various PUFAs ($n = 3$). (B) Western Blotting analysis of PTGIR expression in lymph nodes of mice supplemented with ARA. (C) CD86 expression on activated B cells treated with vehicle, Beraprost, Beraprost plus H89, or 6-Bnz-cAMP alone ($n = 6$). Left: Representative flow cytometry plots. Right: Statistic data of the mean fluorescence intensity (MFI) of CD86. (D) Expression levels of CD86 on stimulated FO B cells following treatment with varying concentrations of Beraprost ($n = 5$). Left: Representative flow cytometry plots. Right: Statistic data MFI of CD86. (E) Flow cytometry analysis of the percentage of CD19$^+$ IgG1$^+$ B cells under stimulations of different concentrations of Beraprost ($n = 3$). Left: Representative flow cytometry plots. Right: Statistic data of the CD19$^+$ IgG1$^+$ B cells. (F) Flow cytometry analysis of GC B cells (B220$^+$ GL7$^+$ FAS$^+$) from mice supplemented with ARA under various inhibitors treatment on day 10 after immunization with OVA ($n = 4$). Left: Representative flow cytometry plots of GC B cells. Right: Statistic data of the percentages and cell numbers of GC B cells. (G, H) B Cell and T cell proliferation measured by CellTrace Violet (CTV) dye in LPS/IL-4-activated murine B cells treated with Beraprost (500 nM) and ARA (1 μM) ($n = 6$). Left: Representative flow cytometry plots. Right: Statistic data of the proliferation index of B cells and T cells. Data are representative of two or three independent experiments. All graphs represent mean ± SEM and all data points represent individual mice. Significance was calculated by one-way ANOVA with Tukey's multiple comparisons test; ns, no statistical significance.

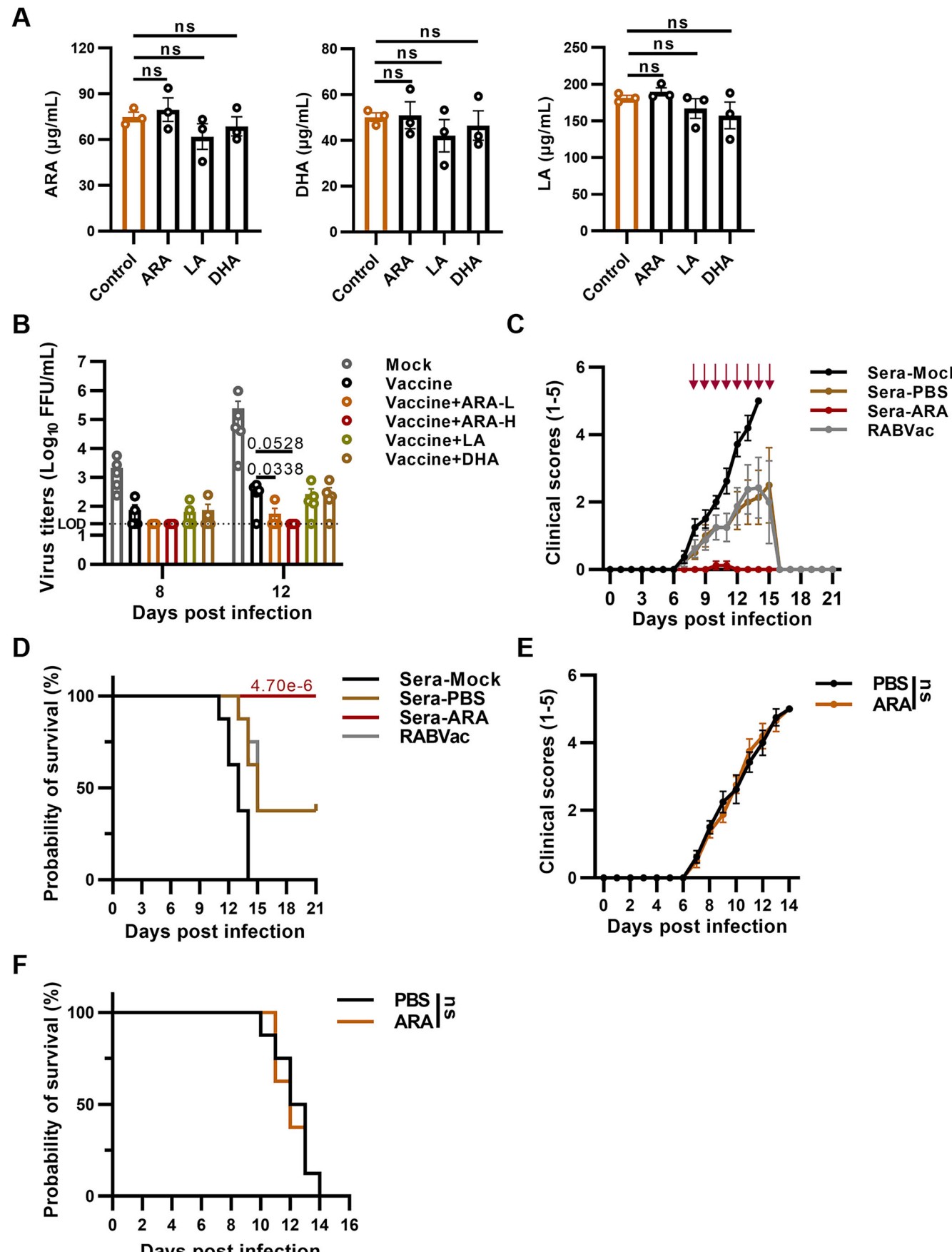

◀ **Figure EV4. Humoral immunity enhanced by ARA supplementation protects immunized mice against RABV challenge.**

(A) Quantitative analysis of ARA, DHA and LA concentrations in murine plasma two weeks following cessation of dietary supplementation ($n = 3$). (B) Virus titers in mouse brain. The mice brains were collected at day 8 and day 12 post-infection and virus titers were calculated and expressed as focus-forming units per ml (FFU/mL) ($n = 5$). LOD, limit of detection. (C, D) Clinical scores and survival curves of mice that received immune sera before challenge ($n = 8$). The mice were intraperitoneally (i.p.) administrated with 200 μL sera. One day later, mice were i.m. challenged with 100 LD50 of RABV. Sera-Mock, the mice were received sera from mice without vaccination. Sera-PBS, the mice were received sera from immunized mice that supplemented with PBS. Sera-ARA, the mice were received sera from immunized mice that supplemented with ARA. RABVac, the mice were immunized with the inactivated rabies vaccine without receiving any sera. (E, F) Clinical scores and survival curves of mice received PBS or ARA ($n = 8$). The mice were administered with ARA (5 mg) or PBS control for 10 days. Fourteen days later, the animals were i.m. challenged with 100 $LD_{50}$ of RABV, and clinical scores and survival were monitored. The arrows (C) indicate a significant difference between the group of Sera-PBS and Sera-ARA. Data are representative of two independent experiments. Data are shown as mean ± SEM and all data points represent individual mice. Significance was calculated by one-way ANOVA with Tukey's multiple comparisons test (A, B), unpaired two-tailed $t$ test (C, E) and log rank (Mantel–Cox) test (D, F); ns, no statistical significance.

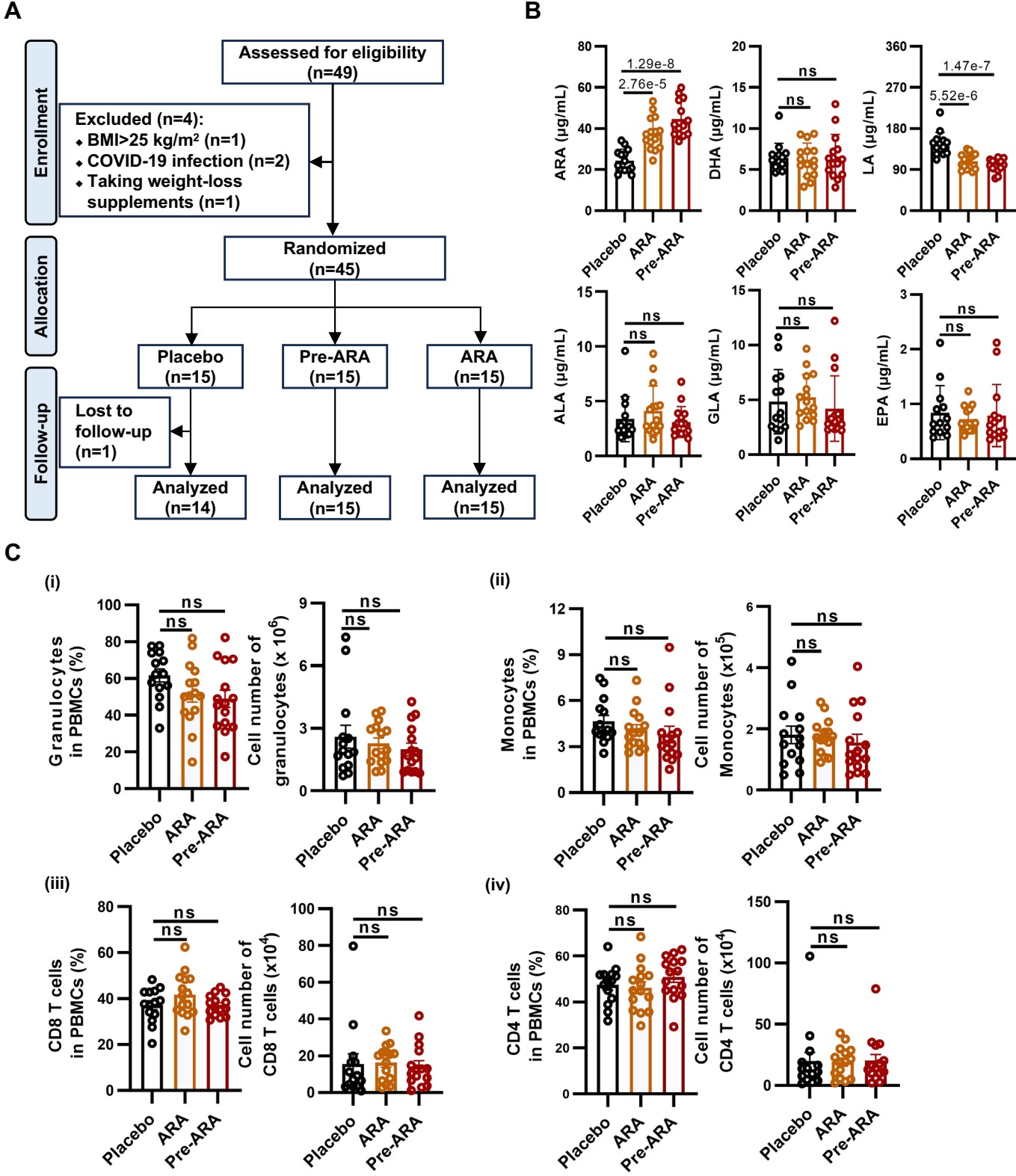

◀ **Figure EV5. The effects of ARA supplementation on PBMCs sourced from volunteers.**

(A) Flow diagram of the study. Placebo, taking 512.4 mg sunflower seed oil daily ($n = 14$). Pre-ARA, taking 512.4 mg of ARA daily on day −3–13 ($n = 15$); ARA, taking 512.4 mg of ARA daily on day 0–13 ($n = 15$); (B) Quantitative analysis of PUFA concentrations in plasma obtained from volunteers supplemented with ARA ($n = 14$ in Placebo group; $n = 15$ in ARA and Pre-ARA group). (C) Statistic data of the percentages and cell numbers of granulocytes, monocytes, total lymphocytes, CD4 T cells, and CD8 T cells in the PBMCs on day 14 after the first shot immunization ($n = 14$ in Placebo group; $n = 15$ in ARA and Pre-ARA group). All graphs represent mean ± SEM, and all data points represent individual volunteers. Significance was calculated by one-way ANOVA with Tukey's multiple comparisons test; ns, no statistical significance.

