## [Peer Review File · EMBO Molecular Medicine]

Dietary supplementation of arachidonic acid promotes humoral immunity

Shengyong Feng, Enhao Ma, Xiaona Na, Zongmei Wang, Wanbo Tai, Xinhui Bao, Mao Wang, Han Chang, Baolei Wu, Miaoqi Liu, Juzhen Li, Huicheng Shi, Celi Yang, Menglu Xi, Haibing Yang, Yuhan Li, Yibin Zhu, Penghua Wang, Ling Zhao, Ai Zhao, and Gong Cheng

Corresponding authors: Ling Zhao (lingzhao@mail.hzau.edu.cn), Ai Zhao (aizhao18@mail.tsinghua.edu.cn), Gong Cheng (gongcheng@mail.tsinghua.edu.cn),

Review Timeline:

Submission Date:	12th Apr 25
Editorial Decision:	20th May 25
Revision Received:	9th Jul 25
Editorial Decision:	14th Aug 25
Revision Received:	28th Aug 25
Accepted:	29th Aug 25

Editor: Zeljko Durdevic

Transaction Report:

20th May 2025

Dear Prof. Cheng,

Thank you for the submission of your manuscript to EMBO Molecular Medicine. We have now received feedback from the two reviewers who agreed to evaluate your manuscript. Both referees recognize interest of the study but also raise important concerns that should be addressed in a major revision. If you would like to discuss further the points raised by the referees, I am available to do so via email or video. Let me know if you are interested in this option.

We would welcome the submission of a revised version within three months for further consideration. Please let us know if you require longer to complete the revision.

I look forward to receiving your revised manuscript.

Yours sincerely,

Zeljko Durdevic

Zeljko Durdevic
Senior Editor
EMBO Molecular Medicine

We require:

- 1) A .docx formatted version of the manuscript text (including legends for main figures, EV figures and tables). Please make sure that the changes are highlighted to be clearly visible.
- 2) Individual production quality figure files as .eps, .tif, .jpg (one file per figure). For guidance, download the 'Figure Guide PDF': (<https://www.embopress.org/page/journal/17574684/authorguide#figureformat>).
- 3) A .docx formatted letter INCLUDING the reviewers' reports and your detailed point-by-point responses to their comments. As part of the EMBO Press transparent editorial process, the point-by-point response is part of the Review Process File (RPF), which will be published alongside your paper.
- 4) A complete author checklist, which you can download from our author guidelines (<https://www.embopress.org/page/journal/17574684/authorguide#submissionofrevisions>). Please insert information in the checklist that is also reflected in the manuscript. The completed author checklist will also be part of the RPF.
- 5) Please note that all corresponding authors are required to supply an ORCID ID for their name upon submission of a revised manuscript.
- 6) It is mandatory to include a 'Data Availability' section after the Materials and Methods. Before submitting your revision, primary datasets produced in this study need to be deposited in an appropriate public database, and the accession numbers and

database listed under 'Data Availability'. Please remember to provide a reviewer password if the datasets are not yet public (see <https://www.embopress.org/page/journal/17574684/authorguide#dataavailability>).

12) Author contributions: You will be asked to provide CRediT (Contributor Role Taxonomy) terms in the submission system. These replace a narrative author contribution section in the manuscript.

13) A Conflict of Interest statement should be provided in the main text.

14) Every published paper now includes a 'Synopsis' to further enhance discoverability. Synopses are displayed on the journal webpage and are freely accessible to all readers. They include a short stand first (maximum of 300 characters, including space) as well as 2-5 one-sentences bullet points that summarizes the paper. Please write the bullet points to summarize the key NEW findings. They should be designed to be complementary to the abstract - i.e. not repeat the same text. We encourage inclusion of key acronyms and quantitative information (maximum of 30 words / bullet point). Please use the passive voice. Please attach these in a separate file or send them by email, we will incorporate them accordingly.

15) Include a Reagents and Tools Table as part of the Methods section, which can be downloaded from our author guidelines (<https://www.embopress.org/page/journal/17574684/authorguide#structuredmethods>)

***** Reviewer's comments *****

Referee #1 (Remarks for Author):

The observation that ARA supplementation enhances virus-specific antibody titers in humans is clinically significant and suggests a potential nutritional enhancement of vaccine efficacy. This finding is well supported by pharmacological studies in mice and ex vivo experiments.

However, it is concerning that the manuscript does not include data on fatty acid profiles in cells, tissues, PBMCs, or blood. Cellular fatty acid composition is critically important for PGI₂ synthesis, as PGI₂ is produced intracellularly by prostaglandin I synthase localized in the ER membrane, in coordination with PLA₂-mediated release of free ARA from membrane phospholipids. The fatty acid composition of cellular membranes is gradually modulated by both the quantity and balance of dietary PUFAs. Both human and murine tissues contain substantial amounts of ARA and other PUFAs, often exceeding the levels introduced through dietary intake.

1. Title

The authors discuss the differences between ARA and LA within the omega-6 PUFAs and highlight the specificity of ARA. The human trial was conducted with ARA and not LA. Therefore, the term "omega-6 polyunsaturated fatty acid" is somewhat misleading. It would be more accurate to refer specifically to "arachidonic acid" in this context.

2. P5 L133-153 Fig 1

2-1

The authors should justify the use of an osmotic pump, as this is an uncommon method for fatty acid administration. It would be helpful to explain the rationale for this approach, particularly in relation to the human trial administration method. Citing previous studies or experiments validating this approach would strengthen the argument.

2-2 To understand whether the 10 mg PUFA dose was appropriate for comparing ARA, LA, GLA, EPA, DHA, and ALA, the fatty compositions of cells, tissues, or serum should be included. The ARA content in cells from the LA or GLA groups may be nearly equivalent to that of the ARA group. Additionally, EPA, DHA, and ALA may significantly reduce ARA levels compared to the control group. Particularly because fatty acid supplementation via an osmotic pump is uncommon, readers may find it challenging to assess the fatty acid profiles of mice given 10 mg PUFAs in this study.

3. P6 L158

Is there the description that ARA intake is 0.1 mg/day in developing countries in ref. 25 or 26? It is too low.

4. P6 L159-214

The term "dietary" is frequently used in the section describing the mice experiments in Fig. 1, but this is misleading because the fatty acids were administered subcutaneously via an osmotic pump. The term should be revised accordingly throughout the manuscript, including in the figures, tables, discussion, and supplementary data.

5. P13 L341

The authors should clarify why LA administration for 3 weeks did not result in increased ARA levels in the cells or improve VNA production. Previous studies indicate that LA is enough metabolized to ARA and further to C-22 PUFAs over such a period. It would be most informative to include data on the fatty acid composition in cells or serum, as suggested in comment 2-2.

6. P14 L378-382

While providing detailed material information is valuable for reader comprehension, the content in Table S4 is related to materials, not results. This information should be relocated to the materials and methods section.

7. P16 L438-446

To support the discussion, essential experimental data are required, as noted in comments 2-2 and 5. In particular, the fatty acid composition of PBMCs or serum from the human trial is critical to interpreting the results.

8. P20 L534

The sources of the PUFAs used in this study should be disclosed (e.g., manufacturer and product details). PUFAs are prone to instability, degradation, and contamination, so providing this information will help demonstrate the appropriateness of the materials and improve the study's reliability.

9. P26 L731

The term "low ARA diet" is unclear. If the authors administered this diet to participants, further details are required. A "low ARA diet" is not commonly discussed in the literature, and it would be challenging to implement without a concurrent low-protein condition.

Table S6 presents only energy and macronutrient content, but the specific amounts of PUFA intake during the intervention period are not provided. This is important, as it would influence ARA and other PUFA compositions in serum and in B and T cells.

Referee #2 (Comments on Novelty/Model System for Author):

High Technical Quality

This cross-species (mouse/human) study tracks ARA conversion to PGI₂ via metabolomics. ARA-PGI₂ activates cAMP-PKA, upregulating CD86/AID to drive immune regulation. A novel *in vivo/vitro* cross-validation system minimizes bias. Clinical monitoring confirms ARA accelerates humoral immunity via rapid antibody elevation, bridging preclinical-clinical gaps and offering translational immune modulation strategies.

High Novelty

This study pioneers ARA (a natural fatty acid) as a novel vaccine adjuvant, contrasting traditional chemical/biological agents. It reveals that ARA-derived PGI₂ activates the cAMP-PKA axis to upregulate CD86 and AID in B cells, identifying a metabolic immune modulation target. Rapid translation from murine models to human trials underscores its translational innovation.

Medical Impact

This study shows ARA, a low-cost dietary supplement, enables single-week protective rabies antibody levels (vs. 3-4 doses), enhancing compliance in resource-limited settings. Its safety and metabolic pathway modulation potential inspire adjuvant redesign for influenza/COVID-19 vaccines, particularly benefiting high-risk populations, and establish dietary-metabolic-immune intervention as a translational paradigm.

Adequate Model System

Mouse and human B cell differentiation and germinal center (GC) response are highly conserved, making them suitable for deciphering molecular mechanisms such as CD86/AID. To validate the efficacy of ARA in healthy volunteers, avoid the limitations of relying solely on animal data, and directly support its medical application value, which provides reference for subsequent clinical trial design.

Referee #2 (Remarks for Author):

In this manuscript, Feng et al elucidates the regulatory role of arachidonic acid (ARA) in adaptive immune responses, particularly through enhancing B cell-mediated antigen-specific antibody production to improve vaccine efficacy. The research demonstrated that dietary ARA supplementation significantly elevated antigen-specific IgG antibody levels compared to other polyunsaturated fatty acids (PUFAs), including LA, GLA, EPA, DHA, and ALA. The mechanism underlying this effect is closely associated with ARA metabolism into eicosanoids. Further investigations revealed that ARA enhances adaptive antibody responses by promoting PGI₂ synthesis, which activates the cAMP-PKA signaling pathway to regulate FO B cell responses.

Notably, the study validated its translational potential through human clinical trials: subjects receiving ARA supplementation exhibited significantly higher specific and virus-neutralizing antibody titers compared to controls. These findings provide a scientific foundation for optimizing vaccine immunization strategies through nutritional interventions.

Overall, the article is well written and easy to understand, and aligns well with the aims and scope of EMBO Molecular Medicine: dedicated science at the interface between clinical research and basic life sciences. It is recommended for publication upon resolution of the following comments :

1. The author does indicate that delivering arachidonic acid to healthy volunteers is safe, but this may not be true for individuals with comorbidities or undergoing immunosuppressive or other treatments. Based on existing literature, evaluate the population heterogeneity challenges that ARA may face as a dietary adjuvant.
2. In fig S2, fig S3, fig S4 and fig S7, although there is no statistically significant difference between the experimental and control

groups, the lack of statistical significance should still be clearly indicated in the figure.

3. L212 the gating strategy of Tfh cells : CD3+CD8-CD4+CD44hiCXCR5+PDI+, PDI should be "PD-1", and please confirm throughout the text.

4. In Fig3B and 4B the first letter of the y-axis title needs to be capitalized, consistent with other figures.

5. L1283-1284 "Mice were immunized with 100 μ L of the solution containing 107 FFU of inactivated rabies vaccine on day...", 107 FFU? please confirm.

6. L1302-1303 "The arrows (D) and (E) indicate a significant difference between the group of ARA-5mg and PBS", "the group of ARA-5mg and PBS", Do they mean vaccine-ARA-H and vaccine group instead? The expression should be consistent.

Dear Editors,

Thank you for the comments on our manuscript (EMM-2025-21805-T) titled **"Dietary supplementation of an Omega-6 polyunsaturated fatty acid promotes humoral immunity"**. We have carefully considered all comments and revised the manuscript accordingly. We are confident that the additional experiments, revised analyses, and textual modifications have significantly enhanced the manuscript.

We were encouraged by the positive comments from the editor and reviewers, such as "Both referees recognize interest of the study" (Editor), "The observation that ARA supplementation enhances virus-specific antibody titers in humans is clinically significant and suggests a potential nutritional enhancement of vaccine efficacy. This finding is well supported by pharmacological studies in mice and ex vivo experiments." (Reviewer #1), "Overall, the article is well written and easy to understand, and aligns well with the aims and scope of EMBO Molecular Medicine: dedicated science at the interface between clinical research and basic life sciences. It is recommended for publication upon resolution of the following comments." (Reviewer #2). The reviewers also provided important suggestions to improve the manuscript. We have incorporated new experimental data into the manuscript based on the reviewers' suggestions. Along with this letter, we provide point-by-point responses to the queries raised by the reviewers.

Responses to Referee #1:

The observation that ARA supplementation enhances virus-specific antibody titers in humans is clinically significant and suggests a potential nutritional enhancement of vaccine efficacy. This finding is well supported by pharmacological studies in mice and ex vivo experiments.

However, it is concerning that the manuscript does not include data on fatty acid profiles in cells, tissues, PBMCs, or blood. Cellular fatty acid composition is critically important for PGI₂ synthesis, as PGI₂ is produced intracellularly by prostaglandin I synthase localized in the ER membrane, in coordination with PLA₂-mediated release of free ARA from membrane phospholipids. The fatty acid composition of cellular membranes is gradually modulated by both the quantity and balance of dietary PUFAs. Both human and murine tissues contain substantial amounts of ARA and other PUFAs, often exceeding the levels introduced through dietary intake.

We thank the reviewer for their insightful comment and we fully agree that characterizing fatty acid profiles in cells, tissues, PBMCs, or blood would strengthen the mechanistic understanding of how ARA supplementation modulates PGI₂ synthesis and antibody responses. Therefore, during the revision, we measured the fatty acid profiles in plasma from both mice and volunteers respectively.

1. Title

The authors discuss the differences between ARA and LA within the omega-6 PUFAs

and highlight the specificity of ARA. The human trial was conducted with ARA and not LA. Therefore, the term "omega-6 polyunsaturated fatty acid" is somewhat misleading. It would be more accurate to refer specifically to "arachidonic acid" in this context.

Thank you for your helpful advice. As instructed, we have updated our title to "Dietary supplementation of arachidonic acid promotes germinal center B cell responses and humoral immunity".

2.P5 L133-153 Fig 1

2-1 The authors should justify the use of an osmotic pump, as this is an uncommon method for fatty acid administration. It would be helpful to explain the rationale for this approach, particularly in relation to the human trial administration method. Citing previous studies or experiments validating this approach would strengthen the argument.

Thank you for your helpful advice. Fatty acids are susceptible to rapid oxidation, enzymatic degradation, and variable absorption following oral administration or bolus injections. Achieving steady-state drug concentrations with conventional oral dosage forms is challenging due to fluctuations in plasma drug levels. This unpredictability poses challenges in attaining targeted concentrations, potentially resulting in either undesirable effects or lack of therapeutic efficacy. Osmotic pumps facilitate continuous, zero-order release kinetics, thereby maintaining stable plasma concentrations and circumventing metabolic degradation pathways [1-3]. Unlike daily gavage or injections, subcutaneous osmotic pumps eliminate stress-induced physiological confounders. Furthermore, this delivery system operates independently of gastrointestinal physiological variables, including motility, pH variation, and food presence [3-5]. Additionally, osmotic pumps are recognized for providing a high degree of in vitro and in vivo correlation [3, 4, 6]. Based on these advantages, we prioritized the subcutaneous osmotic pump administration method to evaluate the biological effects of fatty acids under controlled and physiologically relevant conditions (Line 141-142 Page 6).

v2-2 To understand whether the 10 mg PUFA dose was appropriate for comparing ARA, LA, GLA, EPA, DHA, and ALA, the fatty compositions of cells, tissues, or serum should be included. The ARA content in cells from the LA or GLA groups may be nearly equivalent to that of the ARA group. Additionally, EPA, DHA, and ALA may significantly reduce ARA levels compared to the control group. Particularly because fatty acid supplementation via an osmotic pump is uncommon, readers may find it challenging to assess the fatty acid profiles of mice given 10 mg PUFAs in this study.

Thank you for your helpful advice. To confirm whether 10 mg of PUFAs is appropriate, we conducted additional experiments to quantify PUFA concentrations in murine serum across all experimental groups using LC-MS/MS analysis. The results demonstrated that supplementation with each targeted PUFA significantly elevated its respective concentration in plasma (Fig. EV1A), thus validating the efficacy of this dosage regimen for assessing fatty acid function. However, the supplemented fatty acids generally did not induce significant modifications in other fatty acid levels, with

the exception of a marginal, non-significant decrease in LA levels observed in the group receiving ARA supplementation (Fig. EV1A). Although metabolic conversion from LA/GLA to ARA is possible, some studies suggested this conversion is relatively low [7-9], which may result in the absence of an increase in plasma ARA concentration following supplementation with either LA or GLA. We have added the data to the revised manuscript (Line 155-161, Page 6).

3.P6 L158

Is there the description that ARA intake is 0.1 mg/day in developing countries in ref. 25 or 26? It is too low.

Thank you for pointing out this typo. The actual data for ARA intake are 44–331 mg/day in developing countries and 101–351 mg/day in advanced countries [10]. However, for the lowest-income countries, the per capita median intake of ARA is merely 39 mg/day [11]. In the revised manuscript, we corrected the sentence “ranging from approximately 100 mg/day in developed countries to a mere 0.1 mg/day in developing countries” to “ranging from 101–351 mg/day in developed countries to a mere 39 mg/day in the lowest-income countries.” We have revised in manuscript (Line 166-167, Page 6).

4. P6 L159-214

The term "dietary" is frequently used in the section describing the mice experiments in Fig. 1, but this is misleading because the fatty acids were administered subcutaneously via an osmotic pump. The term should be revised accordingly throughout the manuscript, including in the figures, tables, discussion, and supplementary data.

As indicated in the response to comment #2-1, pump administration offers distinct advantages and was consequently employed for the experiments depicted in Figure 1A-G. This approach minimized confounding effects and enhanced the credibility of our primary findings that ARA enhances the production of antigen-specific antibodies in immunized mice. Nevertheless, oral administration represents a more clinically viable and preferable route for human volunteers compared to subcutaneous osmotic pump implantation. Therefore, we additionally investigated oral administration regimens in mice (Fig. EV1B-C). The results demonstrated that oral ARA supplementation comparably enhances antigen-specific antibody titers. Consequently, oral ARA supplementation was adopted in subsequent experiments, thus the term "dietary" was used.

5.P13 L341

The authors should clarify why LA administration for 3 weeks did not result in increased ARA levels in the cells or improve VNA production. Previous studies indicate that LA is enough metabolized to ARA and further to C-22 PUFAs over such a

period. It would be most informative to include data on the fatty acid composition in cells or serum, as suggested in comment 2-2.

Thank you for your helpful advice. As explained in comment #2-2, LA administration did not significantly elevate plasma ARA concentrations (Fig. EV1A). Furthermore, two weeks post-treatment cessation, all PUFAs concentrations returned to baseline levels (Fig. EV4A). Although linoleic acid (LA) may be metabolized into ARA, the efficiency of this conversion is low [7-9], and exhibits significant variation across different organs [12], potentially attributable to the differential distribution of relevant metabolic enzymes [12, 13]. In this experiment, LA, served as an unrelated control, were supplemented for 10 days (The supplementation lasts from 3 days before immunization to 7 days after immunization) rather than 3 weeks, where "3 weeks" denotes the interval between immunization and viral challenge. Therefore, the amount of LA converted to ARA might be too minimal to elicit a humoral immune response under our experimental conditions.

6. P14 L378-382

While providing detailed material information is valuable for reader comprehension, the content in Table S4 is related to materials, not results. This information should be relocated to the materials and methods section.

Thank you for your helpful advice. As instructed, we relocated the content in Appendix Table S9 to the updated materials and methods section.

7. P16 L438-446

To support the discussion, essential experimental data are required, as noted in comments 2-2 and 5. In particular, the fatty acid composition of PBMCs or serum from the human trial is critical to interpreting the results.

Thank you for your helpful advice. Following the instructions in comments 2-2 and 5, we conducted experiments on the fatty acid composition in the serum of volunteers. The results showed that plasma ARA level was significantly increased in the ARA supplementation group. Except for LA, the levels of other fatty acids remained unchanged after ARA supplementation (Fig. EV5B). Consistent with our study, other studies also found that plasma LA levels were significantly decreased after ARA supplementation, while other fatty acids (e.g. EPA and DHA) remained unchanged [14-16]. We have added the data to the revised manuscript (Line 393-396, Page 14-15).

8. P20 L534

The sources of the PUFAs used in this study should be disclosed (e.g., manufacturer and product details). PUFAs are prone to instability, degradation, and contamination, so providing this information will help demonstrate the appropriateness of the materials and improve the study's reliability.

Thank you for your helpful advice. We have included detailed information on the sources of each PUFAs used in our study in the "Reagents and tools table" section of the updated manuscript.

9. P26 L731

The term "low ARA diet" is unclear. If the authors administered this diet to participants, further details are required. A "low ARA diet" is not commonly discussed in the literature, and it would be challenging to implement without a concurrent low-protein condition.

Thank you for your helpful advice. As per your suggestion, we have revised the term "low ARA diet" to "ARA-restricted diet" throughout the manuscript. Additionally, we have clarified the dietary recommendation used for the ARA-restricted diet (Appendix Table S8).

Table S6 presents only energy and macronutrient content, but the specific amounts of PUFA intake during the intervention period are not provided. This is important, as it would influence ARA and other PUFA compositions in serum and in B and T cells.

Thank you for your valuable suggestions. We fully acknowledge the importance of assessing PUFA intake in this study. As the Chinese food composition table does not provide sufficient data for this purpose, we used the Standard Tables of Food Composition in Japan 2015 (Seventh Edition) to estimate the intake of different types of PUFAs. The results showed no significant differences in the intake of various PUFA types between groups, suggesting that the participants had a generally balanced nutritional intake. We have added these results to Appendix Table S5.

Responses to Referee #2:

Referee #2 (Comments on Novelty/Model System for Author):

High Technical Quality

This cross-species (mouse/human) study tracks ARA conversion to PGI₂ via metabolomics. ARA-PGI₂ activates cAMP-PKA, upregulating CD86/AID to drive immune regulation. A novel in vivo/vitro cross-validation system minimizes bias. Clinical monitoring confirms ARA accelerates humoral immunity via rapid antibody elevation, bridging preclinical-clinical gaps and offering translational immune modulation strategies.

High Novelty

This study pioneers ARA (a natural fatty acid) as a novel vaccine adjuvant, contrasting traditional chemical/biological agents. It reveals that ARA-derived PGI₂ activates the cAMP-PKA axis to upregulate CD86 and AID in B cells, identifying a metabolic immune modulation target. Rapid translation from murine models to human trials underscores its translational innovation.

Medical Impact

This study shows ARA, a low-cost dietary supplement, enables single-week protective rabies antibody levels (vs. 3-4 doses), enhancing compliance in resource-limited settings. Its safety and metabolic pathway modulation potential inspire adjuvant redesign for influenza/COVID-19 vaccines, particularly benefiting high-risk populations, and establish dietary-metabolic-immune intervention as a translational paradigm.

Adequate Model System

Mouse and human B cell differentiation and germinal center (GC) response are highly conserved, making them suitable for deciphering molecular mechanisms such as CD86/AID. To validate the efficacy of ARA in healthy volunteers, avoid the limitations of relying solely on animal data, and directly support its medical application value, which provides reference for subsequent clinical trial design.

Thank you for your valuable feedback regarding our manuscript. We appreciate your recognition of the medical impact of our findings and the technical quality of our experimental design, particularly the model system we constructed.

Referee #2 (Remarks for Author):

In this manuscript, Feng et al elucidates the regulatory role of arachidonic acid (ARA) in adaptive immune responses, particularly through enhancing B cell-mediated antigen-specific antibody production to improve vaccine efficacy. The research demonstrated that dietary ARA supplementation significantly elevated antigen-specific IgG antibody levels compared to other polyunsaturated fatty acids (PUFAs), including LA, GLA, EPA, DHA, and ALA. The mechanism underlying this effect is closely associated with ARA metabolism into eicosanoids. Further investigations revealed that ARA enhances adaptive antibody responses by promoting PGI₂ synthesis, which activates the cAMP-PKA signaling pathway to regulate FO B cell responses.

Notably, the study validated its translational potential through human clinical trials:

subjects receiving ARA supplementation exhibited significantly higher specific and virus-neutralizing antibody titers compared to controls. These findings provide a scientific foundation for optimizing vaccine immunization strategies through nutritional interventions.

Overall, the article is well written and easy to understand, and aligns well with the aims and scope of EMBO Molecular Medicine: dedicated science at the interface between clinical research and basic life sciences. It is recommended for publication upon resolution of the following comments:

1. The author does indicate that delivering arachidonic acid to healthy volunteers is safe, but this may not be true for individuals with comorbidities or undergoing immunosuppressive or other treatments. Based on existing literature, evaluate the population heterogeneity challenges that ARA may face as a dietary adjuvant.

Thank you for the advice. Arachidonic acid is widely present in daily food sources, meaning people would acquire a certain amount of arachidonic acid in their daily lives. In addition, numerous studies have demonstrated that a daily intake of up to 2g of arachidonic acid for 12 weeks is safe for various populations, including children, the elderly, and patients with liver cirrhosis [17]. Since individuals with comorbidities or those undergoing immunosuppressive or other treatments are also permitted to consume meat, poultry, eggs, fish, and other food sources with ARA, the evidence demonstrates the safety of the ARA diet. Nonetheless, studies on the safety of arachidonic acid to patients with comorbidities or immunosuppressive medications are still scant and further study is needed.

2. In fig S2, fig S3, fig S4 and fig S7, although there is no statistically significant difference between the experimental and control groups, the lack of statistical significance should still be clearly indicated in the figure.

Thank you for the advice. We updated those figures by adding the notation "ns" (no statistical significance).

3. L212 the gating strategy of Tfh cells: CD3+CD8-CD4+CD44hiCXCR5+PDI+, PDI should be "PD-1", and please confirm throughout the text.

Thank you for pointing out this typo. As instructed, we corrected "PDI" to "PD-1" in this line and proofread the whole text. We have revised in manuscript (Line 220-221, Page 8).

4. In Fig3B and 4B the first letter of the y-axis title needs to be capitalized, consistent with other figures.

Thank you for pointing out this typo. As instructed, we capitalized the first letter of the y-axis titles of those two figures and double-checked this problem for all other figures.

5. L1283-1284 "Mice were immunized with 100 μ L of the solution containing 10⁷ FFU of inactivated rabies vaccine on day...", 10⁷ FFU? please confirm.

Thank you for pointing out this typo. It should be 10^7 FFU. We corrected that typo and checked for spelling mistakes throughout the text.

6. L1302-1303 "The arrows (D) and (E) indicate a significant difference between the group of ARA-5mg and PBS", "the group of ARA-5mg and PBS", Do they mean vaccine-ARA-H and vaccine group instead? The expression should be consistent.

Thank you for the advice. Yes, "the groups of ARA-5mg and PBS" are indeed the vaccine-ARA-H and vaccine group, respectively. We double-check the naming style of the study groups throughout the manuscript to ensure consistency with the latter style. We have revised in manuscript (Line1173-1174, Page 40).

Reference List

1. Derakhshandeh, K. and M.G. Berenji, *Development and optimization of buspirone oral osmotic pump tablet*. Res Pharm Sci, 2014. **9**(4): p. 233-41.
2. Cheng, X., et al., *Design and evaluation of osmotic pump-based controlled release system of Ambroxol Hydrochloride*. Pharm Dev Technol, 2011. **16**(4): p. 392-9.
3. Almoshari, Y., *Osmotic Pump Drug Delivery Systems-A Comprehensive Review*. Pharmaceuticals (Basel), 2022. **15**(11).
4. Farooqi, S., et al., *Quality by Design (QbD)-Based Numerical and Graphical Optimization Technique for the Development of Osmotic Pump Controlled-Release Metoclopramide HCl Tablets*. Drug Des Devel Ther, 2020. **14**: p. 5217-5234.
5. Verma, R.K., S. Arora, and S. Garg, *Osmotic pumps in drug delivery*. Crit Rev Ther Drug Carrier Syst, 2004. **21**(6): p. 477-520.
6. Sareen, R., N. Jain, and D. Kumar, *An insight to osmotic drug delivery*. Curr Drug Deliv, 2012. **9**(3): p. 285-96.
7. Tallima, H. and R. El Ridi, *Arachidonic acid: Physiological roles and potential health benefits - A review*. J Adv Res, 2018. **11**: p. 33-41.
8. Emken, E.A., R.O. Adlof, and R.M. Gulley, *Dietary linoleic acid influences desaturation and acylation of deuterium-labeled linoleic and linolenic acids in young adult males*. Biochim Biophys Acta, 1994. **1213**(3): p. 277-88.
9. McCloy, U., et al., *A comparison of the metabolism of eighteen-carbon ^{13}C -unsaturated fatty acids in healthy women*. J Lipid Res, 2004. **45**(3): p. 474-85.
10. Kawashima, H., *Intake of arachidonic acid-containing lipids in adult humans: dietary surveys and clinical trials*. Lipids Health Dis, 2019. **18**(1): p. 101.
11. Forsyth, S., S. Gautier, and N. Salem, Jr., *Dietary Intakes of Arachidonic Acid and Docosahexaenoic Acid in Early Life - With a Special Focus on Complementary Feeding in Developing Countries*. Ann Nutr Metab, 2017. **70**(3): p. 217-227.
12. Luthria, D.L. and H. Sprecher, *A comparison of the specific activities of linoleate and arachidonate in liver, heart and kidney phospholipids after feeding rats ethyl linoleate-9,10,12,13-d4*. Biochim Biophys Acta, 1994. **1213**(1): p. 1-4.
13. Cho, H.P., M. Nakamura, and S.D. Clarke, *Cloning, expression, and fatty acid regulation of the human delta-5 desaturase*. J Biol Chem, 1999. **274**(52): p. 37335-9.

14. Kusumoto, A., et al., *Effects of arachidonate-enriched triacylglycerol supplementation on serum fatty acids and platelet aggregation in healthy male subjects with a fish diet*. *Br J Nutr*, 2007. **98**(3): p. 626-35.
15. Kakutani, S., et al., *Supplementation of arachidonic acid-enriched oil increases arachidonic acid contents in plasma phospholipids, but does not increase their metabolites and clinical parameters in Japanese healthy elderly individuals: a randomized controlled study*. *Lipids Health Dis*, 2011. **10**: p. 241.
16. Markworth, J.F., et al., *Arachidonic acid supplementation modulates blood and skeletal muscle lipid profile with no effect on basal inflammation in resistance exercise trained men*. *Prostaglandins Leukot Essent Fatty Acids*, 2018. **128**: p. 74-86.
17. Calder, P.C., et al., *A systematic review of the effects of increasing arachidonic acid intake on PUFA status, metabolism and health-related outcomes in humans*. *Br J Nutr*, 2019. **121**(11): p. 1201-1214.

14th Aug 2025

Dear Prof. Cheng,

Thank you for the submission of your revised manuscript to EMBO Molecular Medicine and please accept my apologies for the delay in getting back to you due to the holiday season. I am pleased to inform you that we will be able to accept your manuscript pending the following final amendments:

- 1) Authors: Please provide institutional email address for the co-corresponding author Ling Zhao and add it on the title page of the manuscript.
- 2) Figures:
 - Main figures and EV figures should be uploaded as individual high-resolution files, with their legends in the manuscript text. Please check "Author Guidelines" for more information:
<https://www.embopress.org/page/journal/17574684/authorguide#figureformat>
<https://www.embopress.org/page/journal/17574684/authorguide#expandedview>
 - It seems that in Figure 3H panels Cerebral cortex/Vaccine and Brain stem/Vaccine are identical. Please check and clarify. Also, make sure that representation of all data in the manuscript is accurate.
 - In Figure EV2A flow cytometry diagram seems to be adopted from a recent publication (<https://doi.org/10.1038/s41598-023-39839-3>). Please reference it in the legend as "Adopted from..." or use an unpublished schematic.
- 3) In the main manuscript file, please do the following:
 - Please address all comments suggested by our data editors listed below:
 - o Figure legends:
 1. Please note that the exact p values are not provided in the legends of figures 1B, D, F, G, H; 2C, E, F, H, J, K; 3B, C, G; 4E, EV1 A, C; EV3 A, C, D, E, F; EV4 D, EV5 B.
 2. Please note that information related to n is missing in the legends of figures 1F, G, H, I; 2K; 4B, C, F, G; EV5 B. S1 A, B; S2B.
 3. Please note that the error bars are not defined in the legends of figures EV4 A, B, C, E.
 - Please correct callouts of Table S5 and S6 to Appendix Table S5 and Appendix Table S6
 - In Methods, provide the statement that informed consent was obtained from all human subjects and confirm that the experiments conformed to the principles set out in the WMA Declaration of Helsinki and the Department of Health and Human Services Belmont Report.
 - In Methods, provide the antibody dilutions that were used for each antibody.
 - Indicate in legends exact n and exact p values, not a range, along with the statistical test used. To keep the figures "clear" some authors found providing an Appendix table Sx with all exact p-values preferable. You are welcome to do this if you want to.
 - Please use the following format to report the accession number of your data:

[data type]: [full name of the resource] [accession number/identifier] ([doi or URL or identifiers.org/DATABASE:ACCESSION])

Please check "Author Guidelines" for more information.

<https://www.embopress.org/page/journal/17574684/authorguide#availabilityofpublishedmaterial>

- 4) Funding: Please enter the information from the "Comments" field in our system in the line-by-line list of funders instead to ensure that the system correctly reflects the complete funding information.
- 5) Source data: Please upload a completed checklist and replace the powerpoint files in the uploaded source data with PDF.
- 6) The Paper Explained: Please add it to the main manuscript text.
- 7) Synopsis:
 - Synopsis image: Please provide the visual abstract as a separate, high-resolution .jpeg file 550 px-wide x 300-600 pixels high.
 - Please check your synopsis text and image before submission with your revised manuscript. Please be aware that in the proof stage minor corrections only are allowed (e.g., typos).
- 8) As part of the EMBO Publications transparent editorial process initiative (see our Editorial at <http://embomolmed.embopress.org/content/2/9/329>), EMBO Molecular Medicine will publish online a Review Process File (RPF) to accompany accepted manuscripts. This file will be published in conjunction with your paper and will include the anonymous referee reports, your point-by-point response and all pertinent correspondence relating to the manuscript. Let us know whether you agree with the publication of the RPF and as here, if you want to remove or not any figures from it prior to publication. Please note that the Authors checklist will be published at the end of the RPF.
- 9) Please provide a point-by-point letter INCLUDING my comments as well as the reviewer's reports and your detailed responses (as Word file).

I look forward to reading a new revised version of your manuscript as soon as possible.

Yours sincerely,

Zeljko Durdevic

Zeljko Durdevic
Senior Editor
EMBO Molecular Medicine

*** Instructions to submit your revised manuscript ***

- 1) a .docx formatted version of the manuscript text (including Figure legends and tables)
 - 2) Separate figure files*
 - 3) supplemental information as Expanded View and/or Appendix. Please carefully check the authors guidelines for formatting Expanded view and Appendix figures and tables at <https://www.embopress.org/page/journal/17574684/authorguide#expandedview>
 - 4) a letter INCLUDING the reviewer's reports and your detailed responses to their comments (as Word file).
 - 5) The paper explained: EMBO Molecular Medicine articles are accompanied by a summary of the articles to emphasize the major findings in the paper and their medical implications for the non-specialist reader. Please provide a draft summary of your article highlighting
 - the medical issue you are addressing,
 - the results obtained and
 - their clinical impact.This may be edited to ensure that readers understand the significance and context of the research. Please refer to any of our published articles for an example.
 - 6) Author contributions: the contribution of every author must be detailed in a separate section.
 - 7) EMBO Molecular Medicine now requires a complete author checklist (<https://www.embopress.org/page/journal/17574684/authorguide>) to be submitted with all revised manuscripts. Please use the checklist as guideline for the sort of information we need WITHIN the manuscript. The checklist should only be filled with page numbers were the information can be found. This is particularly important for animal reporting, antibody dilutions (missing) and exact values and n that should be indicted instead of a range.
 - 8) Every published paper now includes a 'Synopsis' to further enhance discoverability. Synopses are displayed on the journal webpage and are freely accessible to all readers. They include a short stand first (maximum of 300 characters, including space) as well as 2-5 one sentence bullet points that summarise the paper. Please write the bullet points to summarise the key NEW findings. They should be designed to be complementary to the abstract - i.e. not repeat the same text. We encourage inclusion of key acronyms and quantitative information (maximum of 30 words / bullet point). Please use the passive voice. Please attach these in a separate file or send them by email, we will incorporate them accordingly.
- You are also welcome to suggest a striking image or visual abstract to illustrate your article. If you do please provide a jpeg file 550 px-wide x 300-600px high.

9) A Conflict of Interest statement should be provided in the main text

10) Please note that we now mandate that all corresponding authors list an ORCID digital identifier. This takes <90 seconds to complete. We encourage all authors to supply an ORCID identifier, which will be linked to their name for unambiguous name identification.

Currently, our records indicate that the ORCID for your account is 0000-0001-7447-5488.

Link Not Available

11) Include a Reagents and Tools Table as part of the Methods section, which can be downloaded from our author guidelines (<https://www.embopress.org/page/journal/17574684/authorguide#structuredmethods>)

Photos 400-800 DPI

*Additional important information regarding figures and illustrations can be found at

<https://bit.ly/EMBOPressFigurePreparationGuideline>. See also figure legend preparation guidelines:

<https://www.embopress.org/page/journal/17574684/authorguide#figureformat>

***** Reviewer's comments *****

Referee #1 (Remarks for Author):

The paper has been appropriately revised in accordance with the reviewers' comments. The title also accurately describes the content. I believe the paper is worthy of acceptance.

Referee #2 (Remarks for Author):

The authors have satisfactorily addressed all comments of the reviewers and made the requested modifications to the manuscript, leading to significant improvement of the paper, which meets the high-quality standards of EMBO Molecular Medicine. Therefore, I recommend immediate acceptance of the paper for publication in EMBO Molecular Medicine.

The authors addressed the remaining editorial issues.

29th Aug 2025

Dear Prof. Cheng,

We are pleased to inform you that your manuscript is accepted for publication and is now being sent to our publisher to be included in the next available issue of EMBO Molecular Medicine.

Zeljko Durdevic
Senior Editor
EMBO Molecular Medicine
